# Global probabilistic projections of extreme sea levels show intensification of coastal flood hazard

Michalis I. Vousdoukas [1,2], Lorenzo Mentaschi [1], Evangelos Voukouvalas[3], Martin Verlaan[4,5], Svetlana Jevrejeva[6], Luke P. Jackson[7] & Luc Feyen[1]

Global warming is expected to drive increasing extreme sea levels (ESLs) and flood risk along the world's coastlines. In this work we present probabilistic projections of ESLs for the present century taking into consideration changes in mean sea level, tides, wind-waves, and storm surges. Between the year 2000 and 2100 we project a very likely increase of the global average 100-year ESL of 34–76 cm under a moderate-emission-mitigation-policy scenario and of 58–172 cm under a business as usual scenario. Rising ESLs are mostly driven by thermal expansion, followed by contributions from ice mass-loss from glaciers, and ice-sheets in Greenland and Antarctica. Under these scenarios ESL rise would render a large part of the tropics exposed annually to the present-day 100-year event from 2050. By the end of this century this applies to most coastlines around the world, implying unprecedented flood risk levels unless timely adaptation measures are taken.

[1] European Commission, Joint European Research Centre (JRC), Via Enrico Fermi 2749, I-21027 Ispra, Italy. [2] Department of Marine Sciences, University of the Aegean, University hill, 41100 Mitilene, Lesbos, Greece. [3] Engineering Ingegneria Informatica S.p.A. Via S. Martino della Battaglia, 56, 00185 Roma, Italy. [4] Deltares, P.O. Box 1772600 MH Delft, The Netherlands. [5] Department of Mathematics, TU Delft, Van Mourik Broekmanweg 6, Delft 2628 XE, The Netherlands. [6] National Oceanography Centre, Joseph Proudman building, 6 Brownlow Street, Liverpool L3 5DA, UK. [7] Programme for Economic Modelling (EMoD), Nuffield College, 1 New Road, Oxford OX1 1NF, UK. Correspondence and requests for materials should be addressed to M.I.V. (email: Michail.VOUSDOUKAS@ec.europa.eu)

The world's coastline is characterized by high population density, significant socio-economic activity, as well as the presence of critical infrastructure. Coastal areas become threatened when high tides coincide with extreme weather events and drive extreme sea level (ESL) (refs. [1,2] and M.I.V., manuscript submitted). Extreme weather (climate extremes) contributes to ESL through wind-waves and storm surge. Storm surge is an episodic increase in sea level driven by shoreward wind-driven water circulation and atmospheric pressure[3]. Wind-waves are generated when wind energy is transferred to the ocean through surface friction and is transformed into wave energy fluxes[4]. When waves reach the coast they interact with the bathymetry and drive an additional increase in water levels through wave set-up[5] and run up[6]. ESLs are exacerbated by tropical cyclones (TCs), which significantly intensify wind-waves and storm surge[7,8].

Recent findings show that global warming will induce changes in storm surges[7,9] and wind-waves[10,11], while cyclonic activity may be also affected[8,12]. These climate extremes, along with sea-level rise (SLR) will affect ESL and intensify coastal flood risk[1,2,13,14]. Global MSL has been rising during the previous and present century[15] with an accelerated rate[16,17], and is projected to keep doing so for the following decades[18–20]. Rising MSL can affect the phase and amplitude of tides[21,22], as also shown in historical records[23]. Despite these important advances, a coherent global analysis of future ESLs that resolves all the above processes has yet to be conducted. With a few exceptions[13,24], studies on the intensification of future ESLs and the associated rise in flood hazard or risk have only considered SLR, assuming a stationary climate[1,14] and often neglecting wave effects[2,25,26] as well as TCs[3].

Here we combine dynamic simulations of all ESL components during the present century under a moderate-emission-mitigation-policy (RCP4.5) and a high-end, business-as-usual scenario (RCP8.5). We define ESL as a function of mean sea level rise and water levels driven by tides, waves, and storm surges. We used a probabilistic process-based method to calculate regional SLR projections for each RCP[18,27], with SLR projections for RCP 8.5 incorporating larger uncertainties originating from the Greenland and Antarctic ice sheets[28]. We find that the rise in ESLs will result in unprecedented frequency of catastrophic coastal flooding events along many parts of the world. We provide insights on the relative contributions of the different ESL components and their uncertainties.

## Results
### Generation of extreme sea level projections.
We define ESL as

$$ESL = MSL + \eta_{tide} + \eta_{CE}, \qquad (1)$$

where $\eta_{tide}$ is the high tide water level and $\eta_{CE}$ is the water level fluctuations due to climate extremes, i.e., water levels driven by waves and storm surges. Present-day values are obtained from a global reanalysis of tides, wind-waves, and storm surges, including simulations of recorded TCs. Future $\eta_{CE}$ are assessed using wind-wave and storm surge models forced by a 6-member Global Climate Model (GCM) ensemble. Extreme value analysis is applied to the resulting time series to estimate $\eta_{CE}$ for different return periods. We used a probabilistic process-based method to calculate regional SLR projections for each RCP[18,27], with SLR projections for RCP 8.5 incorporating larger uncertainties originating from the Greenland and Antarctic ice sheets[28]. Spatial maps of future MSLs are considered in global tidal simulations to resolve the effect of SLR on high tide water levels $\eta_{tide}$. All ESL components come as probability density functions (PDFs) that express the different sources of uncertainty and that are combined through Monte Carlo simulations in order to generate probabilistic projections of ESLs (Fig. 1). As a metric to understanding potential impacts we focus on changes in the magnitude and frequency of occurrence of the present 100-year ESL, henceforth $ESL_{100}$. Since the study focuses on nearshore ESL dynamics, the global mean values discussed in the manuscript express global coastal averages, omitting the open ocean (see also Methods).

**Global and regional ESL dynamics.** The projected global average $ESL_{100}$ and associated uncertainty ranges progressively increase with time and greenhouse gas forcing (Fig. 2, Supplementary Fig. 2d, Supplementary Figs. 3–5). By the year 2050 we project a very likely increase (5–95th percentile) of the global average $ESL_{100}$ with 14–34 cm and 24–41 cm under RCP4.5 and RCP8.5, respectively (Supplementary Fig. 2d). Differences between the two scenarios are more pronounced toward the end of the century, with the global average $ESL_{100}$ spanning 34–76 cm (RCP4.5) and 58–172 cm (RCP8.5). These global average values mask considerable regional variations. For both RCPs a similar spatial pattern of ESL change is projected, with an increasing trend that is consistent along most of the global coastline. By 2100, the regions with the highest projected ESLs under RCP4.5 are South America (very likely $\Delta ESL_{100}$ of 41–80 cm, Supplementary Table 1), South East Asia (37–79 cm), and South Pacific (29–88 cm). The latter region is projected to experience the highest rise in $ESL_{100}$ under RCP8.5, with a very likely $\Delta ESL_{100}$ of 67–203 cm. Other areas that under RCP8.5 show a rise in $ESL_{100}$ above the global average are Australia (very likely $\Delta ESL$ of 61–179 cm), South America (65–166 cm), South East Asia (62–188 cm) and the west coast of the US and Canada (64–167 cm; Fig. 2, Supplementary Table 1). Locally, areas like the North Sea German coast, as well as parts of East Japan, China, North Vietnam and many of the South Pacific Small Island Developing States are projected to experience the highest increase in the median $ESL_{100}$ exceeding 1 m under RCP8.5 and toward the end of the century. The increase in ESLs is weaker along the coasts of the Baltic Sea, where glacial isostatic adjustment results in a relative sea-level fall that counter-balances and in some cases reverses the rise in MSL and climate extremes[5].

**Contributions from climate extremes.** Projected global average changes in the wind-wave and storm surge ($\eta_{CE}$) component of ESLs show a very weak increasing trend (Supplementary Fig. 2c). The effect of global warming on $\eta_{CE}$ is characterized by high spatial heterogeneity (Fig. 3), with variances that tend to cancel each other when averaged over extensive areas. At regional and local scale, however, the magnitude of changes can be relevant. Along coastlines of the East China Sea and Sea of Japan more intense wind-waves and surges could lift $ESL_{100}$ by up to 30 cm by the end of the century. Under a business-as-usual scenario a large part of the Southern Ocean and North Europe coastlines could experience rises in 100-year $\eta_{CE}$ of nearly 20 cm (Fig. 4, Supplementary Fig. 6; Supplementary Table 3). Projections of future $\eta_{CE}$ show large uncertainty that is more pronounced under a moderate-emission-mitigation-policy scenario, except for Northern Europe. The most prominent decrease is projected along E Africa with the likely range (17–83rd percentile) varying from −32 to −10 cm under a business as usual scenario (Fig. 4, Supplementary Table 3), followed by an extensive domain including Australia, South and South-East Asia (−43 to −2 cm). More moderate decreases in $\eta_{CE}$ (median from −15 to −5 cm) are projected for the north coast of Brazil, most of Central America, and the West Bering Sea (Fig. 3–4, Supplementary Table 3).

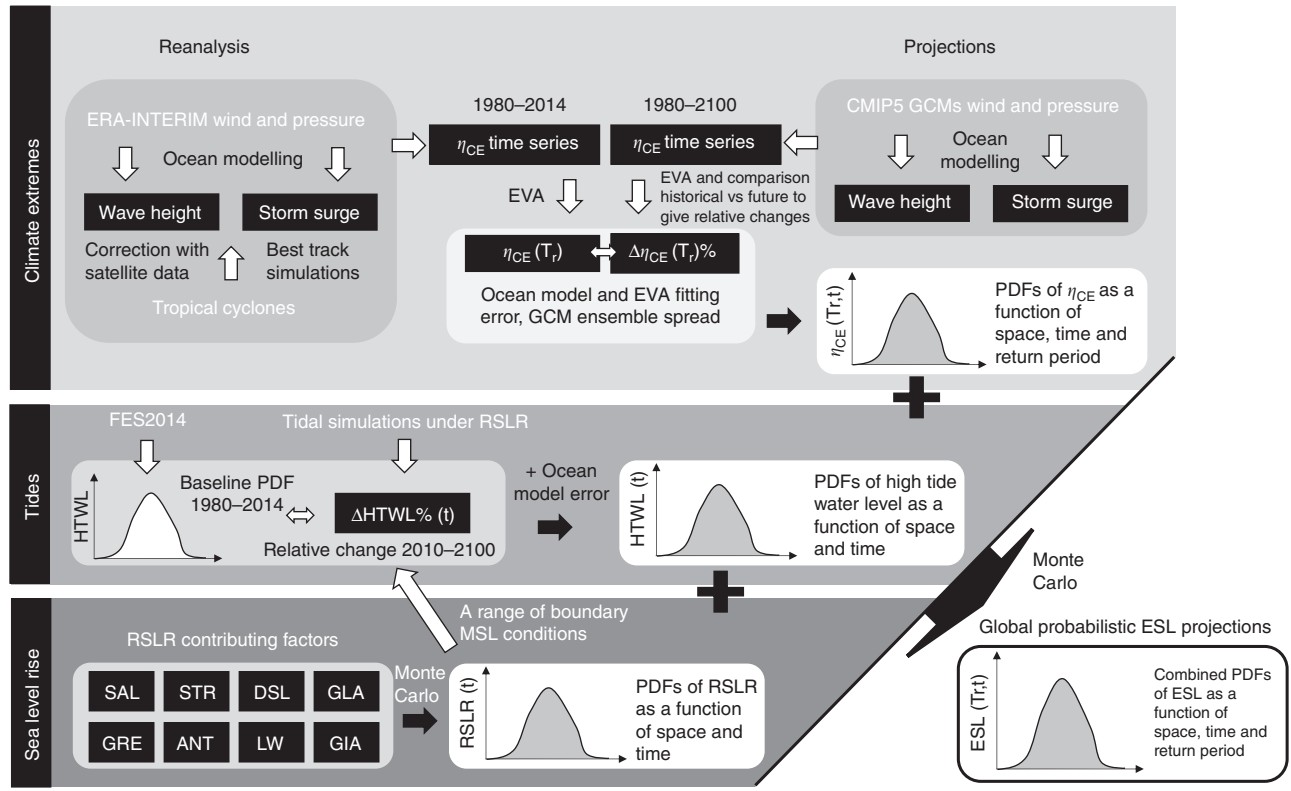

**Fig. 1** Flow diagram showing the procedure to generate the ESL projections. Key: HTWL; high tide water level; $\eta_{CE}$, water level due to climate extremes; PDF, probability density function; EVA, extreme value analysis; RSLR, relative sea-level rise; SAL, self-attraction and loading of the ocean upon itself due to the long term alteration of ocean density changes; STR, steric sea-level change; DSL, dynamic sea-level change; GLA, surface mass balance of ice from glaciers and ice-caps; GRE/ANT, surface mass balance and ice dynamics of Greenland/Antarctic ice sheets; LW, land-water storage and; GIA, Glacial Isostatic Adjustment

Overall there is a tendency for stronger changes toward the end of the century and under RCP8.5, even though there are examples of opposite trends among RCPs or time slices; e.g. along the South Pacific, Alaska, as well as South America at latitudes within 40°–50° (Fig. 4e, f, i). Indices of important teleconnection patterns, such as the North Atlantic Oscillation (NAO), the Antarctic Oscillation (AAO), and the El Niño–Southern Oscillation (ENSO) have been projected to rise in view of climate change[4] and help interpreting most of the projected $\eta_{CE}$ trends. The rise in $\eta_{CE}$ in North Europe shows positive correlation with NAO, while the opposite applies for Central America[29]. ENSO is positively correlated with $\eta_{CE}$ along the Alaska-East Bering Sea and the South Pacific[30] but shows negative coupling with $\eta_{CE}$ for SE Asia, Oceania, Central America, and the West Bering Sea[4]. Finally, positive correlation with the AAO explains the rise of $\eta_{CE}$ along coastlines of the Southern Ocean[4].

**Changes in tides due to SLR**. In line with previous findings[22], local changes in the mean high tide water level ($\eta_{tide}$) are characterized by high spatial heterogeneity (Supplementary Fig. 7); while overall they tend to counterbalance each other at regional or global scales (Supplementary Fig. 2b). The two studied RCPs show contrasting trends, with the exception of Europe, Africa, New Zealand, East China Sea, and the Sea of Japan (Supplementary Fig. 7). As local changes in $\eta_{tide}$ are driven by SLR, they tend to increase with time and are higher under RCP8.5. For the latter scenario a rise of several cm is projected along parts of

North Australia, East Patagonia, and the Sea of Okhotsk (Supplementary Fig. 7). Local decreases of similar magnitude are projected at scattered locations worldwide. However, for most of the world regional changes in tides are insignificant in comparison to the other ESL components; especially under RCP4.5. For the above reasons changes in tides will not be further discussed here, but are provided in the dataset for the convenience of local scale studies.

**Discussion**

Spatial variations in SLR are considerably lower than the global mean trend, which is positive and accelerates with time (Supplementary Fig. 2a). Under the business as usual scenario, the highest SLR toward the end of the century is projected in the South Pacific (median: 95 cm, [very likely range: 54–217]; Supplementary Fig. 8d and Supplementary Table 2), Australia (92 cm, [53–206]), South East Asia (91 cm [52–214]), and Africa (88–89 cm [51–206]). The first 2 areas also rank in the top 2 of the highest projected median rise in $ESL_{100}$, as overall SLR prevails over regional changes in climate extremes, under RCP8.5. However, rising $\eta_{CE}$ in West South America result in the third highest rise in $ESL_{100}$ surpassing South East Asia and Africa. Under RCP4.5, the highest SLR toward the end of the century is projected in the South Pacific (median: 59 cm, [very likely range: 27–97]; Supplementary Fig. 7c and Supplementary Table 2), South East Asia (57 cm [26–93]), Australia (57 cm [25–93]), and West Africa (56 cm [27–89]). The above regions are projected to

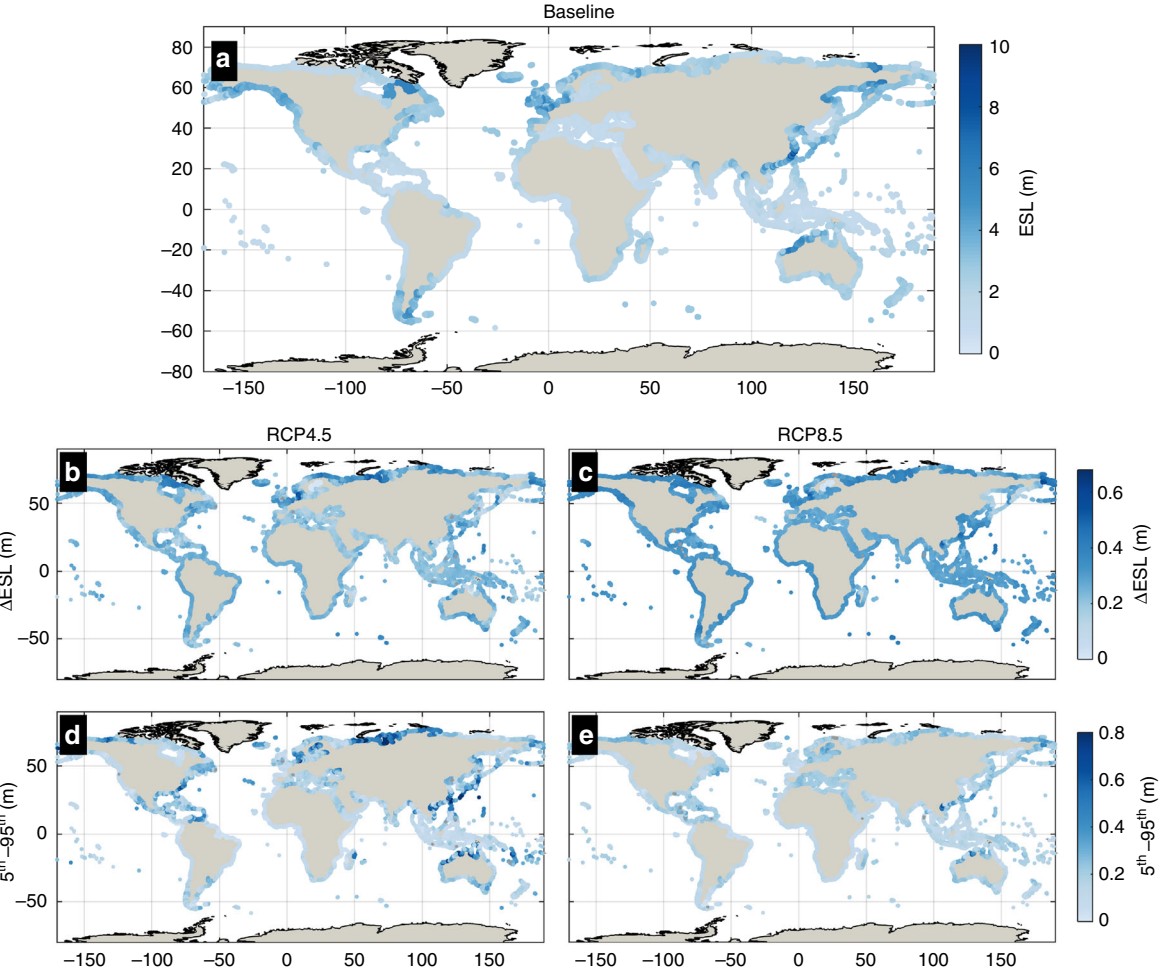

**Fig. 2** Present global ESLs, changes in view of climate change and uncertainty. Maps show the median present-day 100-year ESL (**a**) and the projected changes in $ESL_{100}$ expressed by the median and the very likely range under RCP4.5 (**b**, **d**) and RCP8.5 (**c**, **e**) by 2100

experience a decrease in climate extremes under RCP4.5 (Fig. 3; Supplementary Table 3), and for that reason they are surpassed in the ΔESL ranking by South America (Supplementary Table 1); along which is projected one of the highest rises in $\eta_{CE}$.

A break-down of the projected changes in ESL components shows that steric SLR is the dominant driver of increasing global $ESL_{100}$ during the 21st century (Fig. 5; Supplementary Fig. 9–12). By 2100, it expresses 37% of the global average $\Delta ESL_{100}$ under both RCPs (Fig. 5). The second and third most important contributors vary among RCPs, as melting glaciers are more important than Antarctica (18 vs 12% of $\Delta ESL_{100}$, respectively) under RCP4.5, while under RCP8.5 Antarctica prevails after 2050 (by 2100, Antarctica: 17%; glaciers: 15%). Under both RCPs Greenland contributes 9–10% of $\Delta ESL_{100}$. The effects of dynamic sea level and land-water are the weakest among the SLR components, but aggregated for most regions still outweigh changes in climate extremes after the year 2040. However, it is important to highlight that the geographical domains considered are rather extensive and averaging tends to conceal significant local or regional changes in $\eta_{CE}$ (e.g. see Fig. 3 vs Fig. 4).

Also SLR components have unique spatial patterns related to their physical characteristics—those altering ocean mass (e.g. ice sheets) due to gravitational and rotational effects[31] and those altering volume (e.g. ocean currents). For example, mass loss from Greenland results in a near zero ESL contribution in northwest Europe and eastern Canada, with positive ESL contributions

elsewhere (Figs. 6–7, Supplementary Fig. 9–10). Glaciers contribute negatively to ESL close to their source, particularly Alaska, but positively elsewhere. Antarctica contributes to ESL negatively at the southern tip of South America but positively everywhere, particularly at low-mid latitudes. Land-water contributions are very small and positive along most coastlines, but negative in the Arabian Sea, as well as parts of the Bay of Bengal and the US west coast (Supplementary Fig. 9–10). Along West North America the rising contributions from climate extremes outweigh SLR contributions from land-water, Greenland and dynamic sea level, during most of the century, and under both RCPs (Figs. 6–7, Supplementary Fig. 9–12). In North Europe intensified climate extremes also dominate the effects of glaciers, glacial isostatic adjustment, and dynamic sea level. However, at even smaller scales, changes in climate extremes and tides could gain further importance, dominating most SLR components.

Wahl et al.[2] provide interesting insights into the uncertainties in estimating present and future ESLs, and having expressed all ESL components as PDFs allows to elaborate further on the topic. We express uncertainty as the very likely range (5–95th percentile) and we assess relative contributions from each component to the combined $ESL_{100}$ uncertainty (see also Methods). Present day $ESL_{100}$ uncertainties are related to the predictive skill of the ocean models, as well as the fitting errors during the extreme value analysis of the $\eta_{CE}$ time series. For future estimates, $\eta_{CE}$ uncertainty is increased by the contribution of the inter-GCM

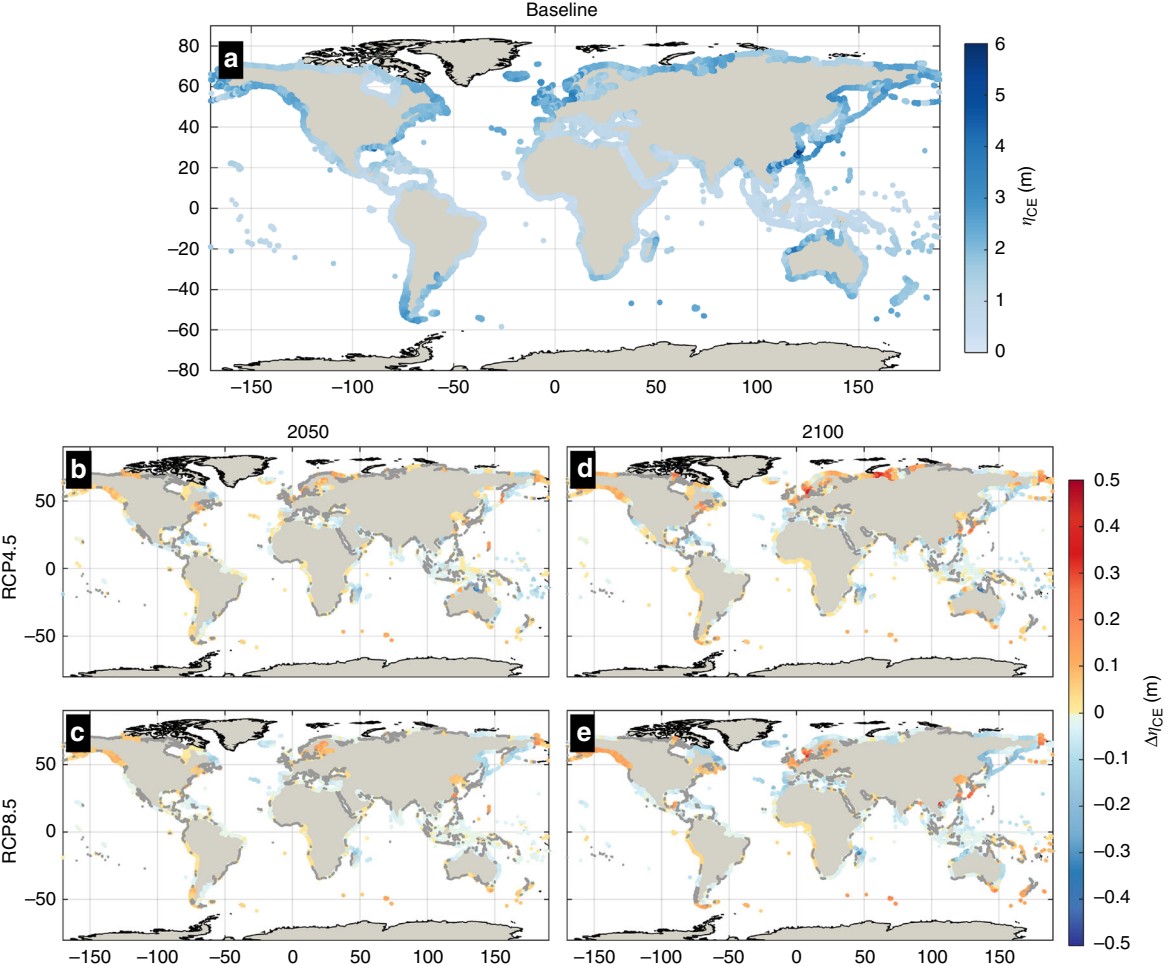

**Fig. 3** Present contributions of climate extremes (wind-waves and storm surges) to global ESLs ($\eta_{CE}$) and projected changes. Maps show the median present-day 100-year $\eta_{CE}$ (**a**) and projected changes ($\Delta\eta_{CE}$) under RCP4.5 by 2050 (**b**) and 2100 (**c**), and under RCP8.5 by 2050 (**d**) and 2100 (**e**). Warm/cold colors express an increase/decrease, respectively, while points with high uncertainty are shown in gray ($|CV| > 1$)

variability with regards to the future climate prediction. Under RCP4.5 and during most of the century, climate extremes remain the main source of uncertainty (Fig. 5c, e), in agreement with previous findings[2]. By the year 2050, 28% of the uncertainty originates from climate extremes, with Antarctica, glaciers and dynamic sea-level change contributing with 15% each. By the end of the century Antarctica contributes 25% of the uncertainty, followed by glaciers and $\eta_{CE}$ (14%). Higher projected SLR ranges under RCP8.5, come with higher uncertainty from the individual components, and dynamic sea-level change is the main source of uncertainty in the near future (Fig. 5d, f). Ice-loss from Antarctica becomes the main source of uncertainty after 2030, with a contribution reaching 50% by the end of the century. Most remaining components have similar contributions ranging from 6 to 10%. To summarize, the upper-tail projections of changes in ESL$_{100}$ under a business as usual scenario are mainly driven by Antarctica ice loss (Fig. 7, Supplementary Fig. 12). Contributions are more balanced under a moderate-emission-mitigation-policy scenario with Antarctica surpassing steric effects only by the end of the century (Fig. 6).

ESL projections are essential for future hazard/risk assessment, coastal planning and the design of coastal protection. Typically, coastal defenses are targeted to withstand ESLs of a certain intensity related to a frequency of occurrence or return period. Translating the projected rise in ESLs into the frequency domain

shows that under both RCPs already by 2050 the present day 100-year event will occur annually in most of the tropics (Fig. 8a, c), rendering many coastal areas exposed to intermittent flood hazard. Such an intensification in frequency is projected for most coastlines around the world by the end of the century, especially under RCP8.5 (Fig. 8b, d; Fig. 9).

The projected intensification of ESLs will likely push existing structures beyond their design limits[32,33]. This will drive an increase in coastal risks, which is already projected to rank very high among natural hazards[34], and has potential to induce massive population movements[26]. Upgrading existing coastal protection would imply increasing elevations by an average of at least 25 cm by 2050 and by more than 50 cm by 2100, but local required increments can be in the order of 1–2 m. Such interventions could have substantial economic, environmental, and societal implications, considering the ~620,000 km of global coastline. All the above highlight the challenging nature of coastal adaptation[35] and the need for timely action toward socially fair and effective strategies.

## Methods

**General concepts.** The models and datasets presented are part of the integrated risk assessment tool LISCoAsT (Large scale Integrated Sea-level and Coastal Assessment Tool) developed by the Joint Research Centre of the European Commission. Extreme Sea Levels (ESL) are driven by the combined effect of the mean sea level (MSL), the high tide water level ($\eta_{tide}$) and water level fluctuations due to

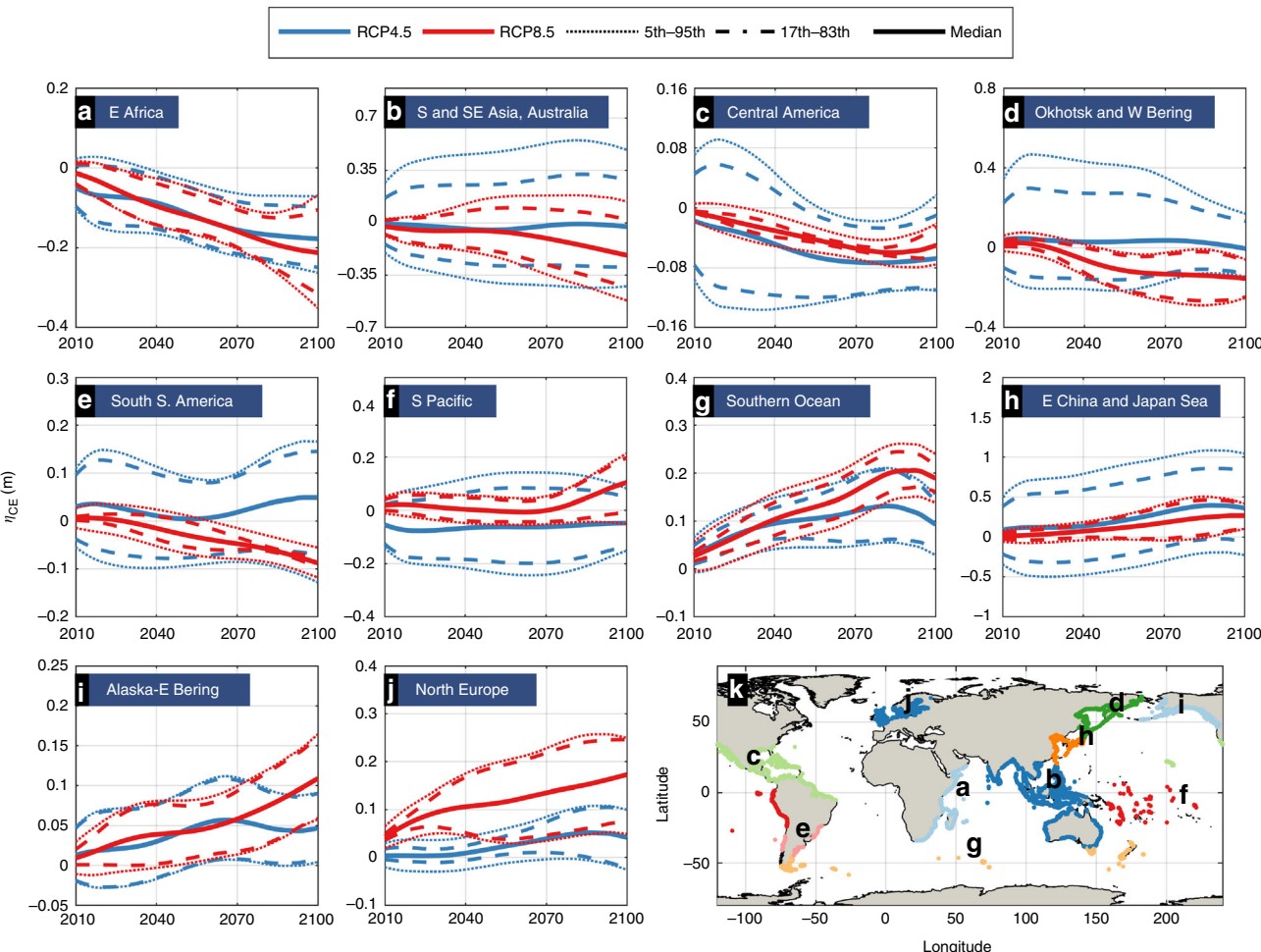

**Fig. 4** Regional projected changes in the contributions of climate extremes (wind-waves and storm surges) to global ESLs ($\eta_{CE}$). Time series of the projected change in 100-year $\eta_{CE}$ under RCP4.5 (blue) and RCP8.5 (red). The geographical regions are highlighted in **k**. Heavy: median, dotted: 5–95th percentiles (very likely) and dashed: 17–83th percentiles (likely)

climate extremes, i.e. water levels driven by waves and storm surges ($\eta_{CE}$). As a result, ESL can be defined as

$$\text{ESL} = \text{MSL} + \eta_{tide} + \eta_{CE} \qquad (2)$$

Projections of physical parameters in view of climate change scenarios are based on two Representative Concentration Pathways (RCPs): RCP4.5 and RCP8.5. RCP4.5 may be viewed as a moderate-emission-mitigation-policy scenario and RCP8.5 as a business-as-usual scenario[36]. The procedure to generate the projections consists of the steps summarized as follows (see also Fig. 1).

Baseline ESL contributions from waves are obtained by a reanalysis, corrected for TC effects based on satellite altimetry data. Similarly, baseline ESL contributions from storm surges are obtained from another reanalysis, and TC contributions are estimated from a third reanalysis simulating all recorded cyclones. $\eta_{CE}$ is estimated by adding the storm surge level ($\eta_{storm\ surge}$) and the wave setup, with the latter approximated as the significant wave height ($H_s$) multiplied by 0.2[5].

$$\eta_{CE} = \eta_{storm\ surge} + 0.2 \cdot H_s \qquad (3)$$

Extreme value analysis provides $\eta_{CE}$ values for different return periods. Following, future $\eta_{CE}$ values are obtained after adjusting the reanalysis values according to the relative changes, estimated from the output of wave and storm surge simulations, forced by climate models covering the period 1980–2100.

Future MSLs are available from probabilistic, process-based projections of regional sea-level change. While present tidal elevations are obtained from available datasets, future changes in tides due to SLR are estimated from simulations considering the range of future MSLs. Finally, all three ESL components come as PDFs, which are combined using a Monte Carlo simulation to generate PDFs of ESLs.

**Baseline values**. The period 1980–2014 is considered as baseline for this study. Present-state tidal elevations along the global coastline are obtained from the FES2014 model (https://www.aviso.altimetry.fr/en/data/products/auxiliary-products/global-tide-fes.html). Given that tidal amplitudes in most places in the world exceed 1 m and can reach 15 m, tides apply an important control on ESLs. Tidal elevations vary during the tidal cycle and typically the duration of extreme events exceeds that of tidal cycles, therefore high tide will occur at least once during a storm. ESLs typically take place when extreme weather coincides with spring tides[37], with the latter occurring twice every lunar month. In order to account for the variability of the tidal elevation, high tide water levels from each tidal cycle are extracted and their PDFs is generated for each point (PDF$_{TIDE,baseline}$).

Hindcasts of waves and storm surges are obtained through dynamic simulations forced by ERA-INTERIM atmospheric conditions, covering the baseline period. Storm surges are simulated using a flexible mesh setup of the DFLOW FM model[3,38]. Waves are simulated using the third generation spectral wave model WW3[4,5]. Both models have been extensively validated and detailed information can be found in the references provided.

Global climate models[39] lack the necessary resolution to fully reproduce the atmospheric fields of TCs, a limitation transferred also to the upper-tail storm surge levels which are usually underestimated by re-analyses such as the present[3]. Therefore we produce an additional reanalysis of cyclone-driven storm surges using the global DFLOW FM setup. All available TC tracks since 1985, found on the Hurricane Research Division of the National and Atmospheric Administration of USA (http://www.aoml.noaa.gov/hrd/hurdat/Data_Storm.html), the Joint Typhoon Warning Center (http://www.usno.navy.mil/NOOC/nmfc-ph/RSS/jtwc/best_tracks/) and the UNISYS database (http://weather.unisys.com/hurricane/) are considered to generate higher resolution atmospheric forcing, using the spiderweb module of the model[40]. Since the domain of the TC simulations is global, technically, each event produces a global dataset of storm surge levels, which are combined with the ones of the ERA-INTERIM reanalysis by selecting the highest value.

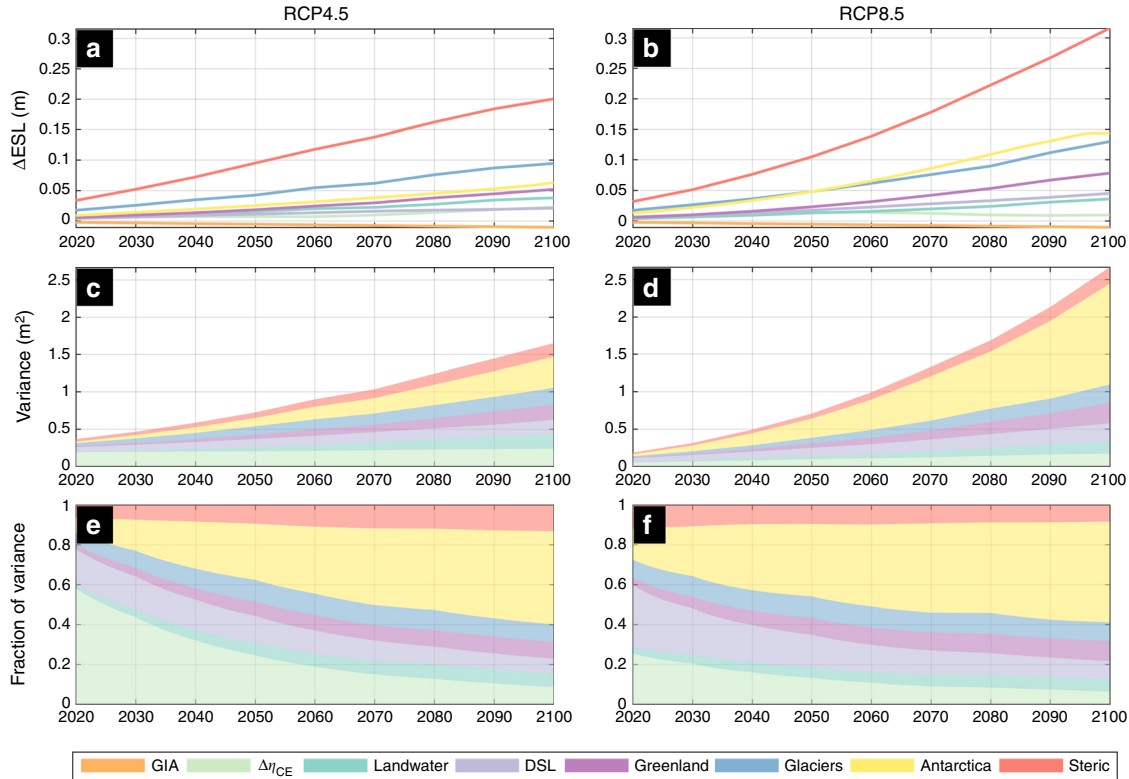

**Fig. 5** Break-down of projected ESL contributions and of their uncertainty. Projected increase of the 100-year ESL from changes in climate extremes, the high tide water level, as well as from SLR contributions from Antarctica, land-water, Greenland, glaciers, dynamic sea level (DSL), glacial isostatic adjustment (GIA), and steric-effects (**a**, **b**); variance (in m²) in components (**c**, **d**) and fraction of components' variance in global ESL change (**e**, **f**); under RCP4.5 (**a**, **c**, **e**) and RCP8.5 (**b**, **d**, **f**). Colors represent different components as in the legend and values express the global mean of the median

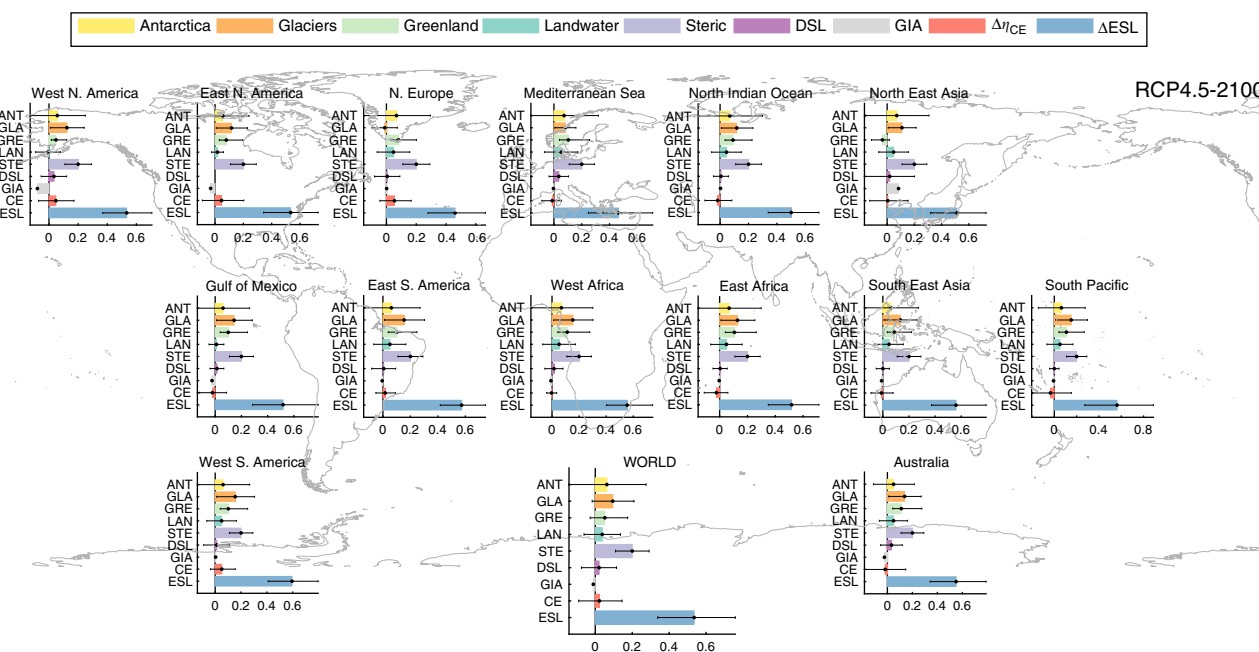

**Fig. 6** Break-down of projected changes in ESL components, under RCP4.5 in 2100. Projected changes in ESL contributions from Antarctica (ANT), glaciers (GLA), Greenland (GRE), land-water (LAN), steric-effects (STE), dynamic sea level (DSL), glacial isostatic adjustment (GIA), climate extremes (CE), as well as the combined 100-year ESL. Bars express the median values, black error lines the 5 and 95 quantiles. Values shown are expressed in m and reflect spatial averages for 14 regions and worldwide

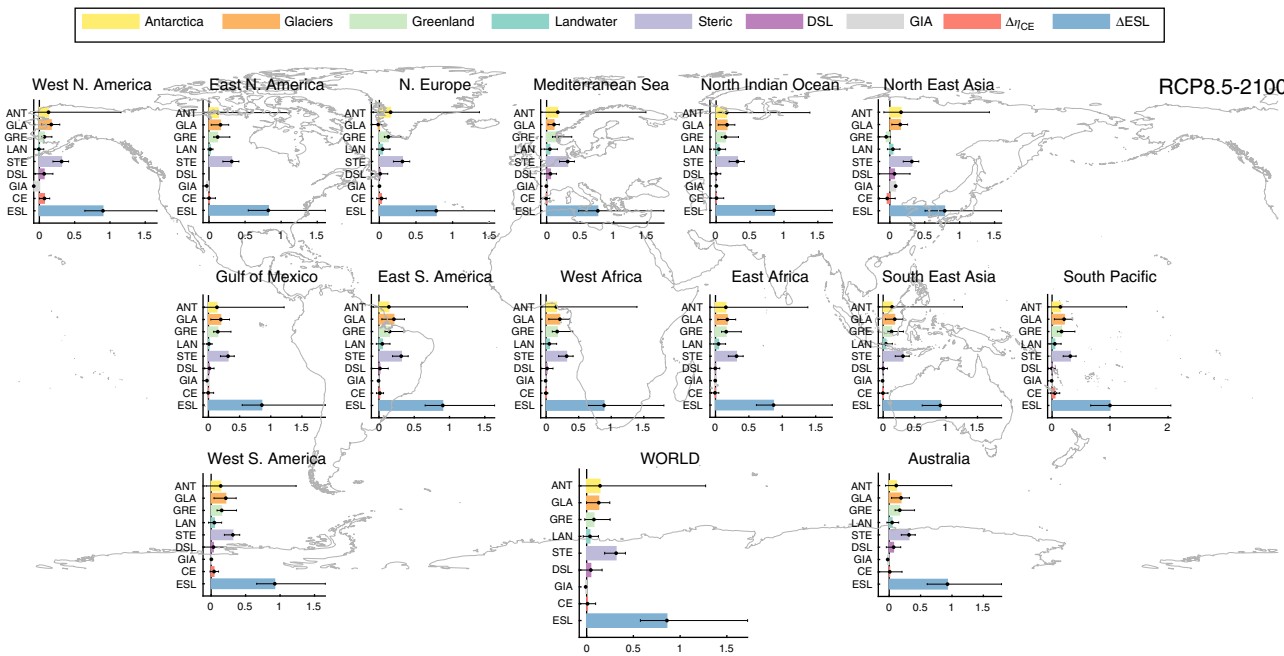

**Fig. 7** Break-down of projected changes in ESL components under RCP8.5 in 2100. Projected changes in ESL contributions from Antarctica (ANT), glaciers (GLA), Greenland (GRE), land-water (LAN), steric-effects (STR), dynamic sea level (DSL), glacial isostatic adjustment (GIA), climate extremes (CE), as well as the combined 100-year ESL. Bars express the median values, black error lines the 5 and 95% quantiles. Values shown are expressed in m and reflect spatial averages for 14 regions and worldwide

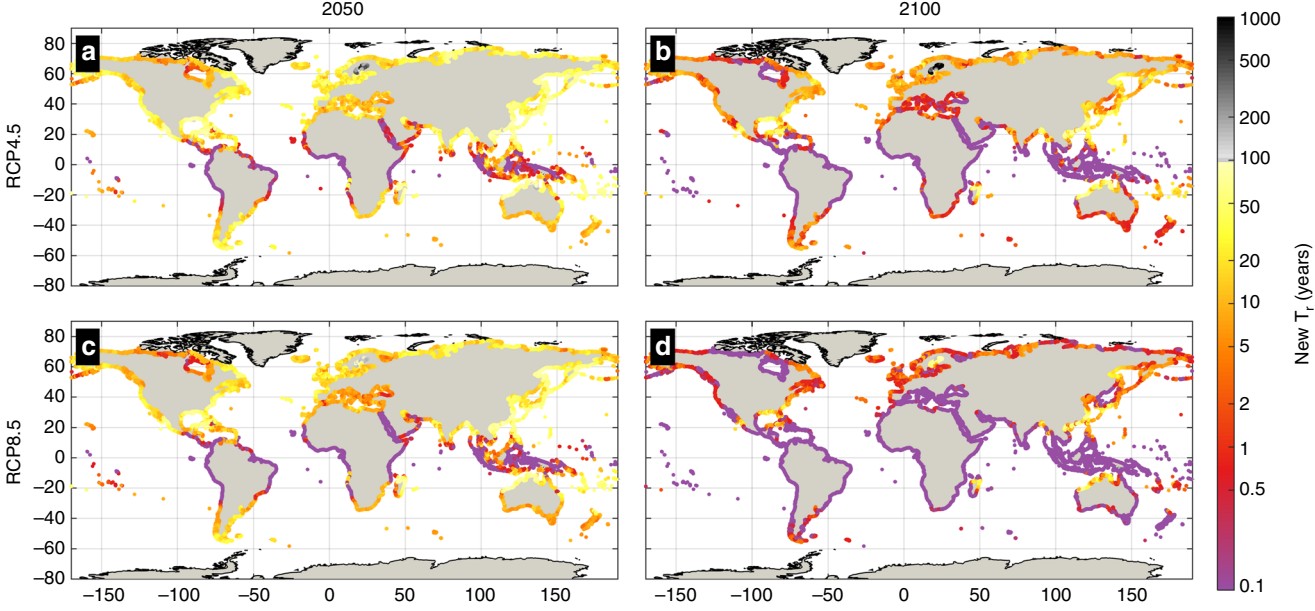

**Fig. 8** Future frequency of the present day 100-year ESL. Colors show the return period of the present day 100-year ESL under RCP4.5 and RCP8.5 in 2050 (**a**, **c**) and 2100 (**b**, **d**), based on the median values. Note that the color scale is not linear

Similarly, we take into account under-predicted TC-driven wave height peaks applying a correction as follows. The TC tracks data (see previous paragraph) and the Globwave satellite altimeter dataset[41] are combined to provide maximum $H_s$ values for each event along coastal locations. More specifically, (i) we estimate maximum values at the intersection of the TC track with the coastline ($H_{s, alt, max}$), considering all altimetry data within 1° distance and a time window of 1. 5 days; and (ii) we replace the reanalysis $H_s$ maxima for the specific period with $H_{s, alt, max}$, in case the former is higher. The above approach is sufficient for correcting the amplitude of the $H_s$ peaks, which is the information used in the extreme value analysis to follow.

The resulting storm surge level and $H_s$ time series are combined to generate $\eta_{CE}$ time series according to Eq. (2) and $\eta_{CE}$ values for different return periods are obtained using a stationary version of a non-stationary extreme value analysis package[42]. Even though the 100-year return period is discussed in the manuscript, the dataset includes also data for the 5, 10, 20, 50, 200, 500, 1000 year events.

**Projections of ESLs in view of climate change.** SLR projections. We use a probabilistic, process-based approach to project relative, sea-level rise (RSLR) up to the end of this century[27]. To the first order one can sum individual sea-level components to give total sea-level change that on the regional scale can be written

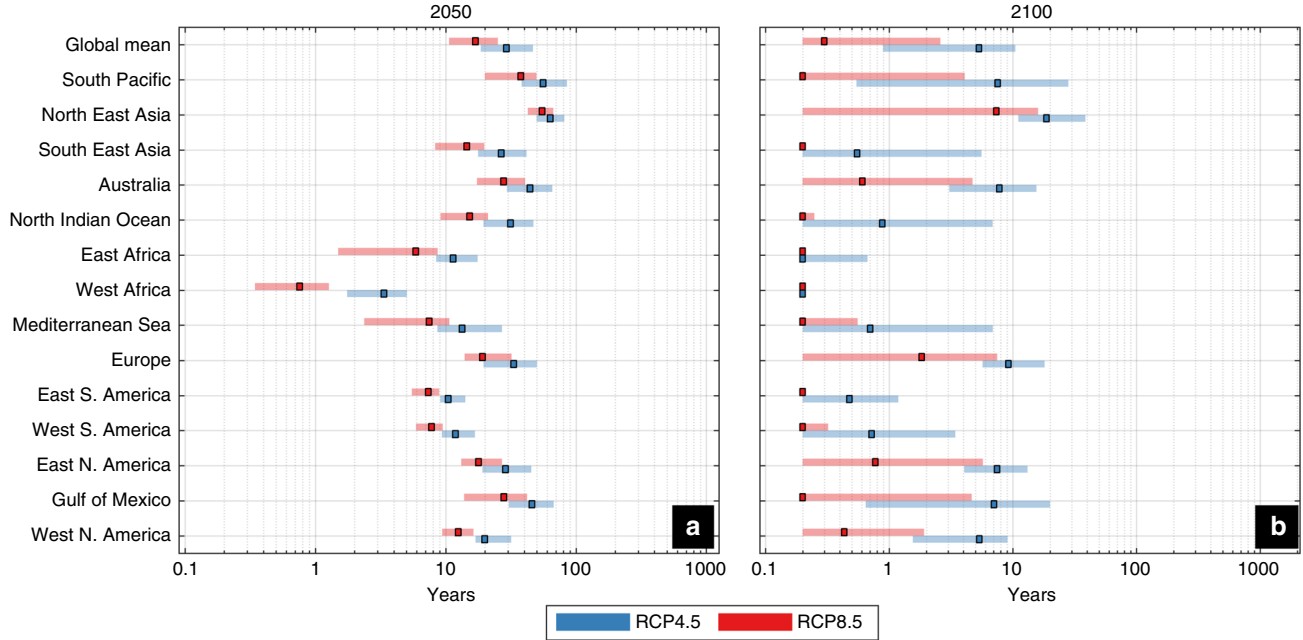

**Fig. 9** Future frequency of the present day 100-year ESL along 14 geographical regions. Return period of the present day 100-year ESL under RCP4.5 and RCP8.5 in 2050 (**a**) and 2100 (**b**). Colored boxes express the median and colored patches the 5 and 95% quartiles. The values shown are averages along the global coastline as well as along the coasts of 14 geographical regions

as the sum of each components time dependent global average projection (e.g. GRE $(t)$) multiplied by its associated fingerprint (e.g. $F_{GRE}(\theta, \varphi)$),

$$
\begin{aligned}
RSLR(\theta, \varphi, t) = \quad & F_{SAL}(\theta, \varphi) \cdot [STR(t) + DSL(\theta, \varphi, t)] \\
& + F_{GLA}(\theta, \varphi) \cdot GLA(t) \\
& + F_{GRE}(\theta, \varphi) \cdot GRE(t) \\
& + F_{ANT}(\theta, \varphi) \cdot ANT(t) \\
& + F_{LW}(\theta, \varphi) \cdot LW(t) + GIA(\theta, \varphi) \cdot t.
\end{aligned} \tag{4}
$$

The contributions in Eq. (3) are the impact of self-attraction and loading (SAL) of the ocean upon itself due to the long term alteration of ocean density changes, globally averaged steric sea-level change (STR), dynamic sea-level change (DSL), surface mass balance of ice from glaciers and ice-caps (GLA), surface mass balance and ice dynamics of Greenland (GRE) and Antarctic (ANT) ice sheet, land-water storage (LW) and Glacial Isostatic Adjustment (GIA).

Each projected global average component is represented by a PDF at each time slice. We randomly sample each PDF 5000 times at each time slice and scale each realization by its associated fingerprint. We sum one realization from each component and repeat this for all realizations to give 5000 realizations of total projected RSLR at each time slice for each scenario. Next, we add the time-integrated, scenario independent global field of sea-level change due to GIA[43] multiplied by the relative time difference. We then estimate the combined PDF and calculate quantiles from these realizations at each grid-point to give probabilistic regional sea-level change. The provenance of each sea-level component, its associated fingerprint and time dependent global projection (including PDFs) are described in Jackson and Jevrejeva[27]. For RCP8.5, we assume a high-end RSLR projection[18,27], which uses Greenland and Antarctic ice sheet contributions by Bamber and Aspinall[28]. The result of this change is a higher median global sea-level change than the conventional RCP8.5 projection[44] with uncertainties that are larger and asymmetric.

**Projections of tidal elevation, storm surges and waves**. The effect of SLR on global tidal elevations is assessed through a set of simulations, using the global DFLOW FM set-up. Time-varying spatial maps of the RSLR (updated every decade) are considered to modify the MSL in the simulations and high tide water levels $\eta_{tide}$ are extracted. The above are compared with $\eta_{tide}$ values from simulations for the baseline period and the relative changes ($\Delta \eta_{tide}\%$) are estimated along the

coastline, considering 10-year windows:

$$
\Delta \eta_{tide}\% = 100 \times \frac{\eta_{tide,projected} - \eta_{tide,baseline}}{\eta_{tide,baseline}}. \tag{5}
$$

The simulations are repeated for both RCPs and each of the following RSLR quantiles (5th, 17th, 50th, 83rd, 99th) resulting in PDFs of $\Delta \eta_{tide}\%$ (PDF$_{\Delta TIDE\%}$). Baseline $\eta_{tide}$ values are adjusted according to the $\Delta \eta_{tide}\%$:

$$
\eta_{tide,projected} = \left(1 + \frac{\Delta \eta_{tide}\%}{100}\right) \times \eta_{tide,baseline}. \tag{6}
$$

Eq. (4) is applied in a probabilistic approach through Monte Carlo simulations sampling from 3 PDFs: PDF$_{TIDE,baseline}$, PDF$_{\Delta TIDE\%}$, and a PDF$_{model\ uncertainty}$ expressing the ocean model forecasting error. PDF$_{model\ uncertainty}$ is assumed to be a Gaussian with mean equal to zero and standard deviation equal to 10% of the $\eta_{tide}$, as model validation[45] showed normalized mean square errors around 10%. During the Monte Carlo simulations PDFs are discretized in 100 percentiles and random sampling includes at least 100,000 values and stops only when a convergence criterion for the quantiles is satisfied. The final result is PDFs of the future $\eta_{tide}$ every 10-years under both RCPs, and along the global coastline.

Atmospheric forcing from a 6-member CMIP5 Global Climate Model (GCM) ensemble[46] is used to obtain projections of future waves and storm surges as well as their changes in relation to the historical period (1980–2014). Using the same set-up as for the baseline simulations, waves are obtained with the WW3 model and storm surges using DFLOW FM. This resulted in time series of climate extremes-driven water level variations $\eta_{CE}$, generated for each GCM for the period 1980–2100. Since future cyclone tracks are not available, projections of climate extremes were generated with the assumption that trends in $\eta_{CE}$ estimated from the GCMs, express also relative changes in $\eta_{CE}$ including TCs. The above assumption is justified by the fact that, even though not represented in detail, cyclonic structures are present in the GCM atmospheric fields, therefore changes in cyclonic activity leave their footprint in the wave and storm surge simulations.

Following, non-stationary extreme value statistical analysis[42] is applied to obtain $\eta_{CE}$ values for different return periods. Similar to $\eta_{tide}$, the final $\eta_{CE}$ projections are obtained after adjusting the reanalysis values according to the relative changes obtained from the CMIP5 simulations (equation 5). Again we apply a probabilistic approach combining in Monte Carlo simulations the PDFs from the following sources of uncertainty: the $\Delta \eta_{CE}\%$ GCM ensemble spread, as well as the errors from the extreme value distribution fit and the ocean model. Further information about the CMIP5 GCM ensemble, the model setup and validation can be found in the literature cited[4,5].

**Spatial analysis**. In this study we consider ~5000 points distributed along the global coastline every ~100 km. The global coastline is divided in 14 geographical regions in order to identify regional patterns in the ESL component trends. For the spatial analysis of changes in climate extremes additional, more confined regions are defined on the basis of criteria such as geographical proximity, and similarities in metocean, atmospheric conditions, as well as in trends in climate extremes. All values discussed in the manuscript correspond to averages either for each region, or for the entire global coastline.

**Statistical analysis**. The PDFs of all ESL components are available along the global coastline, for each considered RCP, return period and at 10-year time steps, until the end of the century. The individual components' PDFs are considered as independent parameters and ESL PDFs of their joint contributions are obtained through Monte Carlo simulations following the steps below[18,27]: (i) random sampling from the individual PDFs (MSL, $\eta_{tide}$, $\eta_{CE}$); (ii) linear addition of each realization of the ESL components according to Eq. (1); (iii) control of convergence to ensure that the number of realizations is sufficient; (iv) combined PDF estimation. Typically one million realizations are sufficient to obtain satisfactory convergence in the PDFs and the final percentiles. The result is ESL PDFs that express the contributions from all components and the different sources of uncertainty: GCM inter-model variability, range of the different SLR contributions, ocean models predictive skill, extreme value distribution fitting errors, footprint of changing sea levels on tidal amplitudes.

The fact that all ESL components are available as PDFs allows one to evaluate their relative contributions both to the ESLs and to the combined uncertainty. The most important components are analyzed: $\eta_{tide}$, $\eta_{CE}$, as well as contributions from Antarctica, land-water, Greenland, glaciers, and steric-effects. The relative contribution of a component to the change in ESLs is expressed by the fraction of its median change to the total median change. Similarly, relative contributions to the total ESL uncertainty are expressed by the fraction of each component's variance to the total variance.

The coefficient of variation CV is used to address the agreement of the estimated trends among GCMs and projected changes with $|CV| > 1$ (i.e., 84% probability) are considered as uncertain and are thus omitted[5]. The statistical significance of the projected changes is assessed through Mann–Kendall and only significant changes are discussed in the manuscript.

**Limitations**. This study does not address non-linear interactions between SLR components, tidal flows, waves, and storm surges; thus we treat all ESL components as independent variables. Such an assumption can affect the accuracy of the results since factors contributing to MSL changes show dependence[47], and so do the other studied ESL components. The water depth modulates the bottom friction and the water extent and thus water level variations are important both for nearshore wave processes[48] and storm surges[49,50]. In addition, tidal and wind-driven currents interact[51] and can also affect wave fields through Doppler effects[52]. The choice of treating all components as independent can be justified by the combination of the following: (i) resolving all the non-linear effects would require a fully coupled modeling approach, which however goes beyond the current modelling and computational capabilities; (ii) non-linear interactions have been shown to be important locally[49,52], but previous studies have demonstrated the validity of the assumption for climate change projections[53–56], as the resulting error is outweighed from other sources of uncertainty and therefore, the approach is common in similar large-scale studies[57]; and (iii) the current assumption allows quantifying better the uncertainties in each ESL component and their combination, a very essential aspect in studies on climate change projections. Further discussion on the topic can be found in previous related studies[5,45].

Coastal sea-level will also be strongly dependent upon local land motion. We have incorporated the only globally predictable, continuous spatial field of land motion, glacial isostatic adjustment[43] (GIA). However, at smaller scales local tectonics and anthropogenically induced subsidence (e.g. ground water pumping) have the potential to completely dwarf other sea-level (and hence extreme sea-level) components[58]. On the other hand, working at the ~100 km scale probably averages out much of the potential local land subsidence leaving the large wavelength GIA signal as the main factor.

The skill of ocean models to reproduce tides, waves and storm surges[59–61] has been shown to be affected by the spatial and temporal resolution of the atmospheric forcing and the computational grid. The presently considered spatio-temporal scales reflect a practical limit in the ocean modelling resolution that may affect the models' performance in locations with complex morphology. Validation efforts have produced satisfactory results; however, in this aspect the work can benefit by future progress in computational resources and modelling capabilities. Moreover, near the poles, beyond latitudes 60° N or 60° S, ice content is not taken into account by the wave and storm surge models, even though it can affect oceanic mass and energy fluxes. Therefore, the results along these areas are less reliable. Similarly, coastal recession can reverse the changes in tides under SLR[21,22], but for practical reasons we have assumed a constant shoreline in this study. All the above can be important for future ESLs as tides and climate extremes drive episodic increases in sea levels of several meters, and therefore local changes even up to 5–10% can leave a notable footprint in terms of risks.

Wave contributions to ESLs are expressed by a, previously applied, generic approximation of wave setup (refs. [5,62] and M.I.V., manuscript submitted). The calculation is based on off-shore wave parameters and assumes the same slope of the shoaling zone all along the global coastline. The approach is simpler than in previous smaller-scale studies[1,6], but the above assumptions are inevitable given the lack of nearshore topographic data at the resolution needed to resolve more accurately wave shoaling and breaking patterns. Similarly, ESL contributions from swash motions[63] are omitted, since despite their importance[64], their estimation requires knowledge of the beach-face slope[65,66] which is largely unknown at global scale.

GCMs lack the resolution to reproduce the atmospheric fields of TCs, despite the fact that the latter are important drivers of ESLs and coastal risk in many parts of the world. In this study, all recorded cyclones have been simulated and considered, for the first time, in a global reanalysis of ESLs. A potential extension of the work would be to consider a large set of synthetic tracks[13] in order to resolve the full spectrum of probabilities of a cyclone affecting a stretch of coastline. Such analysis could be very challenging at global scale and for that reason previous studies have taken place only on regional/local scales[7,13,67]; while most large-scale studies do not consider cyclone effects[3]. Furthermore, in our analysis we assume that the CMIP5 GCM resolution which is lower than the one of the TC reanalysis simulations, does not distort the projected changes in $\eta_{CE}$. Overall, all assumptions are acceptable given the current state of the art and the scope of the study and the reader can find further discussion on the methodological aspects in previously published research[5,45].

**Code availability**. The Delft3D-FM code is currently being made available in http://oss.deltares.nl. The WW3 code is available in http://polar.ncep.noaa.gov/waves/wavewatch/license.shtml. The code applied for the non-stationary extreme value statistics[42] available in https://github.com/menta78/tsEva.

**Data availability**. The global ESL data that support the findings of this study are available in the LISCoAsT repository of the JRC data collection (http://data.jrc.ec.europa.eu/collection/LISCOAST) though this link: http://data.jrc.ec.europa.eu/dataset/jrc-liscoast-10012, with the identifiers: https://doi.org/10.2905/jrc-liscoast-10012; PID: http://data.europa.eu/89h/jrc-liscoast-10012.

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

## Acknowledgements

The research leading to these results has received funding from the EU Seventh Framework Program FP7/2007–2013 under grant agreement no 603864 (HELIX: "High-End cLimate Impacts and eXtremes"; www.helixclimate.eu). Support from CAPES for Special Visiting Professor (PVE) under the project number 88881.068343/2014–01 is also acknowledged.

## Author contributions

M.I.V., and L.F. jointly conceived the study. L.J., and S.J. produced the SLR projections, M.I.V., M.V., and E.V. contributed with the storm surge and tidal elevation projections;

and L.M., and M.I.V. were responsible for the wave projections. M.I.V. analyzed the data and prepared the manuscript, with all authors discussing results and implications and commenting on the manuscript at all stages.

## Additional information

**Competing interests:** The authors declare no competing interests.

