## [Peer Review File · Nature Communications]

Reviewers' comments:

Reviewer #1 (Remarks to the Author):

This manuscript assesses changes in extreme sea levels during the 21st century along the world coastlines. Extremes, as defined here, are caused by the combined effect of mean sea level rise, the associated changes in the tidal range and high waters due to storm surges and wave setup induced by wind-waves. The authors use different sources to project the contributors to extreme sea levels under two RCP scenarios and then combine the corresponding individual probabilities into a single one. To do so, a probabilistic approach has been followed. They conclude that extreme sea levels will increase significantly along most of the global coastlines, with the main driver being the ocean thermal expansion on average.

The present manuscript is well organised and well written, although the quality of some of the figures should be improved. All the details on the data origin and processing are provided. The analyses performed are sound and support the conclusions of the work. The authors include a section on the limitations of their approaches in the Supplementary Information; these reflect mostly the inherent limitations of numerical models as well as some necessary simplifications of a global study. Overall I think this work deserves publication, but I would first encourage the authors to address the issues that I am listing below. I think one important limitation of their approach, in particular the way the contributors to extreme sea levels are combined, is not accounted for and should be discussed. Here below I am including a list of comments, sorted by relevance that I hope will help to improve the manuscript.

Major comments:

- In my opinion, one major issue is the way in which the pdfs are combined. The pdfs of the individual contributors are assumed to be independent, but in reality they are not: MSL rise impact directly on tidal ranges, for example. And changes in GHGs that drive MSL rise also produce significant surge variations at the regional scales. This means that not all combinations of the pdfs are equally likely. I am not suggesting that the authors should repeat the work accounting for these inter-dependencies (I do not even have the clue of how to address it!), but I think that this should be clearly pointed out and discussed in the manuscript. My gut feeling is that, globally averaged, the impact would be small; however, in regions where surge and wave extremes suffer significant (positive/negative) changes the results would be biased (low/high).

- One single PDF is computed for the combined effect of surges and wind-waves, as indicated in eq. 2 in the Supplementary Information. This is because both numerical models are forced with the same atmospheric fields and so their outputs are consistent. This is fine when extremes from surges and waves are generated by the same storms and are thus correlated. But it may not be that accurate in places where the coasts are threatened by remote swells rather than by local wind-seas. In this case, regional differences in models may lead to a random combination of surges and Hs and I think it would be more appropriate to combine the two individual pdfs.

Moderate comments:

- Figure 1 (and possible Figure 2, too) would be more informative if the percentage of change with respect to present-day values is included. A large change may not be that significant in regions that are already used to large extremes.

- The TC hindcast: this is a very intuitive and nice way to include TC in surge and waves hindcasts. Of course, TC are not included in the projections (this is actually mentioned in the section on limitations), so I disagree with the following statement "Since future cyclone tracks are not available, projections of climate extremes were generated with the assumption that trends in η CE estimated from the GCMs, express also relative changes in η CE including TCs. The above assumption is justified by the fact that, even though not represented in detail, cyclonic structures are present in the GCM atmospheric fields, therefore changes in cyclonic activity leave their

footprint in the wave and storm surge simulations.” This justification does not hold, as one could say the same for present-day climate. This makes me wonder that, if these events are not included in the projections, what is the point of having TC in the baseline hindcast. Would it be more correct to simply not accounting at all for TC, since the changes are not in the projections anyway? I think the authors need to either justify their choice or remove the TC from their estimates.

Minor comments:

- The quality of the figures showing the maps should be improved. For example, in figure 2 and S6 it is hard to distinguish the colours along the coastlines.

- The term global average used in the manuscript (e.g. last paragraph in page 2, figure S3) refers to global coastal average. This is clear when reading the supplementary but it would be good to make a statement also in the main text. The confusion may arise easily, especially for some components such as MSL which are often discussed in terms of global ocean values.

- Define SIDS (page 3)

- “Regional variations in SLR are considerably lower than the global mean trend,” (page 5): In absolute terms, regional/local changes cannot be overall smaller than their average. Do you refer to regional deviations with respect to global MSL? Is this the global ocean or the global coastline? Please clarify.

- Page 7: figures 10-12-> figures S10-S12

- “Upgrading existing coastal protection would imply increasing elevations by an average of at least 25 cm by 2050 and by more than 50 cm by 2100”: personally I do not like this statement because it oversimplifies the results and is not representative of the impacts. As the authors themselves claim, the local values may reach 1-2 m, and I am sure that we agree that these are the important numbers.

- “On the other hand, working at...” (page 9, SI): I don’t understand the sense of this sentence. Why it refers only to uplift and not to subsidence? Whatever the case, it is in contradiction with the sentence just above.

Reviewer #2 (Remarks to the Author):

Reviewer: Sean Vitousek

The paper presents a global-scale estimation of changes in extreme water levels, which contribute to coastal flooding driven by regional sea-level change, waves, tides, and storm surge. The authors examine, for the first time, the effect of non-stationary climate change on the wave, tide, and storm surge patterns on extreme coastal water levels, which should make an excellent addition to the literature. Although, there a few notable regions which are greatly affected by a climatic changes in storminess, such as the North Atlantic and North Pacific. Many regions are altered insignificantly due to changes in storminess under the RCP 4.5 and 8.5 emission scenarios considered here, and instead are dominated by sea-level rise (SLR), suggesting that earlier works that consider stationary wave climates (Vitousek et al., 2017 – Scientific Reports) may suffice for large-scale coastal hazard assessments.

I was considering writing a paper such as the current manuscript, which considers changes in storminess; however I was glad to see that the authors beat me to it, because I would have not such a thorough analysis as the work presented here. Personally, I am somewhat skeptical of the robustness and utility of non-stationary projections of extreme storminess. Especially, since

recent work (Wahl et al., 2017 – Nature Communications) has demonstrated that the uncertainties in extreme water levels can in some cases exceed that of SLR, and therefore trying to estimate non-stationary changes on top of this can be difficult to constrain. Never-the-less, the authors of the current manuscript seem to have gone above and beyond the call of duty in terms of using their modeling efforts to estimate possible changes in storminess, despite their findings of relatively small changes, overall. Ultimately, their findings reinforce the critical role that SLR plays in future hazard projections, as well as a few locations where changes in storminess are important.

Overall, I enjoyed reading the manuscript and I recommend only a few minor revisions.

Minor revisions:

1) Figure 2: There seems to be some strange behavior between the RCP 4.5 and RCP 8.5 scenarios near western Russia and west Australia where the tendency switches from a positive to a negative contribution from RCP 4.5 to 8.5. I was under the impression that going from 4.5 to 8.5 should increase the tendency, not reverse it. Is there an explanation of what is going on here? Does the fundamental stochastic nature of the climatic extremes realized in these models make this type of analysis difficult?

2) Figure 2: This figure shows the absolute change in extreme water level ($\Delta\eta_{ce}$). A large number of these data points are less than 10-15 cm, while many of the extreme water levels exceed 3 m, suggesting that the percent changes are very small. I personally would be very interested in the percent changes in extreme water level. For example, I may be helpful to have a supplemental figure showing the percent change in extreme water level? Or what percent of locations show less than 5% change, what areas show greater than 10% change. Based on Wahl et al., (2017), I have little doubt that the uncertainty in the stationary extreme water level (in most cases) will be much larger than the non-stationarity. Never-the-less, it is still an important thing to investigate, like the authors have done. Could the uncertainty estimates like in Wahl et al., (2017) be used to estimate the significance of the changes in storminess reported here?

3) Equation 2: $Runup = 0.2 H_s$. In my opinion, this is quite a bit of an over-simplification, since the wave driven-water level (e.g., setup/runup) will certainly depend on wavelength (or wave period), as in the Stockdon (2006) empirical equation. I see that the authors have used this formulation in a previous report, so it is indeed a published methodology. However, if possible, it may be helpful to examine the influence of wavelength (or wave period), especially to complement the VERY robust approach taken for all of the other water level components.

4) Page 5: Local changes in the mean high tide water level: I think the discussion on the changes in the tide due to SLR should be shortened even further in the main text, as in my opinion it would be completely insignificant in comparison to the other factors (and uncertainties) affecting extreme water level. If I were conducting such a study, I would have guessed that the effect of SLR on the astronomic tide would be completely insignificant (perhaps except in shallow estuaries), and thus I never would have conducted such an analysis. However, I commend the authors' diligence to actually verify this for the rest of us.

5) Figure 3: I am a bit unclear on what is meant by variance and the sources of it here. I would appreciate some additional discussion and examples on how this is estimated. (As a side note: Why is the variance for $\Delta\eta_{ce}$ constant for RCP4.5? but not for RCP 8.5?)

6) Figures 1,2,5: In our paper (Vitousek et al., 2017), we presented our global-scale estimates of coastal hazards using continuous color plots (like in Figure S9) rather than at discrete points along the coastline. This approach seemed more intuitive to us since, we were considering continuous oceanic variables (despite their coastal relevance), and the clipping of those variables to coastal points seemed somewhat arbitrary and furthermore it may result in some under representation of small island nations. I am not suggesting that the authors change this approach, however, I wonder if they have a compelling reason/motivation for keeping the results only near

the (major) coastlines? We debated these different approaches with our reviewers before ultimately sticking with the continuous maps.

7) Figures S5 and S13: I really like these analyses and figures a lot. To me, they are more informative than Figures 2 and 5, respectively. Thus, I believe they may be worth migrating to the main article rather than in the supplemental materials. However, I am happy with whatever the authors/editors decide.

Reviewer #3 (Remarks to the Author):

This is an interesting paper that deals with an important and timely topic. The authors employ a probabilistic framework to assess future changes in the different components of extreme sea levels (i.e. mean sea level, tides, storm surges, and waves). Their approach allows the authors to quantify the contributions of individual drivers and the associated spatial variability. I think the presented results are significant and relevant to the broad readership of Nature Communications. However, there are a few aspects that need more attention/revision, before the manuscript can be recommended for publication.

General comments:

- A paper was recently published by Rasmussen et al. "Extreme sea level implications of 1.5 °C, 2.0 °C, and 2.5 °C temperature stabilization targets in the 21st and 22nd century", <https://doi.org/10.1088/1748-9326/aaac87>. The paper was probably not available at the time the authors submitted their manuscript, but it is now, and since it deals with a similar topic it would be important to discuss the results from this present manuscript in light of the results of this other study. I believe that this manuscript is different enough and has new significant findings still warranting its publication through a high-level outlet.
- A bit more explanation for using $0.2*H_s$ as an approximation for the wave setup would be interesting. In their paper, Vitousek et al. (2017) use a slightly different approach. It would be important to understand how sensitive the results are to different wave setup approximations.
- In the supplementary material I was a bit confused by the discussion on the importance of TCs and how they are incorporated. First, for the historical period, the authors correctly state that TC storm surges are likely underestimated when using only the reanalysis data as boundary conditions, and hence decided to include observed TC information from various data bases as additional forcing. Then, for future simulations, it is stated on page 6 that relative changes in TC storm surges should be captured by the GCMs. Further down, on page 10, the fact that GCMs poorly resolve TCs is then listed as a limiting factor in the studies. I think this is a crucial aspect and needs more/better explanation and consistency.

Detailed comments:

- In the current version the authors introduce their definition of ESL at the beginning of the supplementary material (i.e. sum of MSL, tide, surge, wave setup). Given that the entire study is built around this definition it should also appear prominently in the main text.
- P.2 first para. Saying that only "low-latitude" coastal zones are affected by tropical cyclones is a bit misleading and ill-defined. I am thinking about events travelling up the US east coast affecting areas well beyond 35 or 40 degrees latitude.
- P4. It might be clearer to say decrease and then use -10 to -32 cm. Technically it's not wrong how it is done now, but I was a bit puzzled first.
- P.7 It should be "Figures S10-S12"

Reply to reviewers' comments

Authors: We would like to thank the reviewers for their comments, which allowed us to improve the manuscript. We have prepared a revised version which also confronts with the formatting of the journal. Apart from styling changes we have shortened the Abstract to 150 words, moved from the SI to the main manuscript the Methods, as well as some figures following the reviewers' suggestions. We have also submitted the figures in high quality .eps format, but we left the lower quality ones inside the main manuscript to facilitate the reviewers.

Reviewer 1

This manuscript assesses changes in extreme sea levels during the 21st century along the world coastlines. Extremes, as defined here, are caused by the combined effect of mean sea level rise, the associated changes in the tidal range and high waters due to storm surges and wave setup induced by wind-waves. The authors use different sources to project the contributors to extreme sea levels under two RCP scenarios and then combine the corresponding individual probabilities into a single one. To do so, a probabilistic approach has been followed. They conclude that extreme sea levels will increase significantly along most of the global coastlines, with the main driver being the ocean thermal expansion on average.

The present manuscript is well organised and well written, although the quality of some of the figures should be improved. All the details on the data origin and processing are provided. The analyses performed are sound and support the conclusions of the work. The authors include a section on the limitations of their approaches in the Supplementary Information; these reflect mostly the inherent limitations of numerical models as well as some necessary simplifications of a global study. Overall I think this work deserves publication, but I would first encourage the authors to address the issues that I am listing below. I think one important limitation of their approach, in particular the way the contributors to extreme sea levels are combined, is not accounted for and should be discussed. Here below I am including a list of comments, sorted by relevance that I hope will help to improve the manuscript.

Authors: We would like to thank the reviewer for his comments, which we found that were spot on and have helped us improve the manuscript.

Major comments:

1) In my opinion, one major issue is the way in which the pdfs are combined. The pdfs of the individual contributors are assumed to be independent, but in reality they are not: MSL rise impact directly on tidal ranges, for example. And changes in GHGs that drive MSL rise also produce significant surge variations at the regional scales. This means that not all combinations of the pdfs are equally likely. I am not suggesting that the authors should repeat the work accounting for these interdependencies (I do not even have the clue of how to address it!), but I think that this should be clearly pointed out and discussed in the manuscript. My gut feeling is that, globally averaged, the impact would be small; however, in regions where surge and

wave extremes suffer significant (positive/negative) changes the results would be biased (low/high).

Authors: We agree with the reviewer that the non-linear interactions are not properly resolved in our analysis and the reason is that as he/she points out there is no feasible way to do it (at least in our understanding). In our initial submission we were discussing this point in the Limitations section of the SI. In the revision the Methods are included in the main manuscript; while at the same time we expanded the discussion to address better our treatment of the non-linear effects. So we think that this point is more transparent for the reader.

2) One single PDF is computed for the combined effect of surges and wind-waves, as indicated in eq. 2 in the Supplementary Information. This is because both numerical models are forced with the same atmospheric fields and so their outputs are consistent. This is fine when extremes from surges and waves are generated by the same storms and are thus correlated. But it may not be that accurate in places where the coasts are threatened by remote swells rather than by local wind-seas. In this case, regional differences in models may lead to a random combination of surges and Hs and I think it would be more appropriate to combine the two individual pdfs.

Authors: We are not really convinced that the suggested change would improve the analysis. The reviewer is right in saying that in some locations waves (W) and storm surges (SS) are correlated and in others not. The way we combine the two components allows us to capture both situations. We hope that the reviewer agrees that in the areas where there is correlation the current approach is superior, since by combining the PDFs (and not the series) we would be treating two dependent variables as independent. We agree with the reviewer that combining the PDFs at the locations where there is no correlation between W and SS could be an interesting direction to explore for one main reason: In this case W and SS are driven by intense meteorological conditions occurring at different areas and even times and the timing of the meteorological extremes in the CMIP5 GCMs is stochastic (e.g. different ensemble numbers could potentially result in different combinations of W and SS). So which approach is best depends on how common is the dependence between the two variables, and this is definitely an interesting direction to explore at global scale, but probably would go beyond the scope of the present study. Findings from a European study show that the two variables are more often dependent (at least to some extent) than independent (for example see Figure 10 in <https://www.nat-hazards-earth-syst-sci-discuss.net/nhess-2017-177/nhess-2017-177.pdf>); which supports the use of the current approach. Also for the areas with low dependence the fact that we use 6 GCMs adds some randomness in the way the two variables are combined.

Moderate comments:

3) Figure 1 (and possible Figure 2, too) would be more informative if the percentage of change with respect to present-day values is included. A large change may not be that significant in regions that are already used to large extremes.

Authors: We agree with the reviewer, but we also think that preference to relative or absolute changes depends on personal taste or the message one tries to convey (see also our reply to comment 2 from Reviewer 2). In our initial submission we decided to discuss changes in the frequency domain (i.e. return periods), as we considered them more intuitive for coastal planners, given that most coastal construction projects are designed on that basis. In the revision we have included additional variants of (now) Fig 2 and 3 with the relative change in the SI.

4) The TC hindcast: this is a very intuitive and nice way to include TC in surge and waves hindcasts. Of course, TC are not included in the projections (this is actually mentioned in the section on limitations), so I disagree with the following statement “Since future cyclone tracks are not available, projections of climate extremes were generated with the assumption that trends in η_{CE} estimated from the GCMs, express also relative changes in η_{CE} including TCs. The above assumption is justified by the fact that, even though not represented in detail, cyclonic structures are present in the GCM atmospheric fields, therefore changes in cyclonic activity leave their footprint in the wave and storm surge simulations.” This justification does not hold, as one could say the same for present-day climate. This makes me wonder that, if these events are not included in the projections, what is the point of having TC in the baseline hindcast. Would it be more correct to simply not accounting at all for TC, since the changes are not in the projections anyway? I think the authors need to either justify their choice or remove the TC from their estimates.

Authors: The comment is fair as dealing with TCs in such large-scale projections is delicate. It is also relevant to a comment from Reviewer 3. We did our best to explain our reasoning but we can understand that some aspects are still not well conveyed. TCs are present in the GCM atmospheric fields, but they are not resolved in detail, as the low GCM resolution applies a smoothing effect. This translates into an underestimation of the storm surge levels, however it doesn't mean that TCs are absent from the GCM fields, they are just roughly represented. In the high-resolution TC reanalysis this issue is resolved with the spiderweb grid following the TC track and resolving in detail the cyclonic pattern.

We believe that having a good reanalysis including TCs is already an asset for the work since the baseline values are an improvement from the current state of the art, and we are confident that the community will use the dataset. The critical point is whether the relative change estimated from the CMIP5 simulations applies also to the TC reanalysis. Basically the validity of our assumption comes down to effect of atmospheric forcing resolution to the estimated storm surge relative changes. This is still a topic which has not been thoroughly studied but results from current research tell us that the model resolution does not apply strong effect on the estimated relative changes in view of climate change (see figure below where the patterns of change between the first and second column, expressing different atmospheric forcing resolutions, are very similar). Overall we cannot claim that our assumption is 100% solid, but it is the best approximation given the present capabilities and we believe that it introduces errors which are overshadowed by other sources of uncertainty. Still we acknowledge that the way we presented our

treatment of TCs was not optimal and we have improved the manuscript to that direction, by expanding and clarifying the discussion.

Figure 1 Effect of the atmospheric resolution on projected changes in significant wave heights between the baseline and the period 2070-2100 under RCP8.5. Absolute and relative change (C_{abs} and $C\%$) of the 99th H_s percentile estimated for atmospheric forcing with resolution 0.75° (a and d respectively) and 1.5° (b and e). In both cases the GCM is EC-EARTH and waves are model with WAVEWATCH-II at 0.4° grid resolution. Panels c and f show the absolute difference of C_{abs} and $C\%$, respectively, obtained from the two different resolution atmospheric resolutions.

Minor comments:

5) The quality of the figures showing the maps should be improved. For example, in figure 2 and S6 it is hard to distinguish the colours along the coastlines.

Authors: The figures provided in the pdf for the review process are lower resolution than the original ones which were produced in vector form and are sharper. In general we will make sure that the final figures will be up to the standards.

6) The term global average used in the manuscript (e.g. last paragraph in page 2, figure S3) refers to global coastal average. This is clear when reading the supplementary but it would be good to make a statement also in the main text. The confusion may arise easily, especially for some components such as MSL which are often discussed in terms of global ocean values.

Authors: In the revision we harmonize the manuscript to the journal's style and we clarify the issue raised by the reviewer in the last sentence of the introduction:

'Since the study focusses on nearshore ESL dynamics, the global mean values discussed in the manuscript express global coastal averages, omitting the open ocean (more details on the different steps of the analysis can be found in Supplementary Methods).'

7) Define SIDS (page 3)

Authors: We have modified the manuscript accordingly.

8) "Regional variations in SLR are considerably lower than the global mean trend," (page 5): In absolute terms, regional/local changes cannot be overall smaller than

their average. Do you refer to regional deviations with respect to global MSL? Is this the global ocean or the global coastline? Please clarify.

Authors: We are referring to the change in the global mean MSL during the century that is higher than the observed spatial MSL variability at any given time. This applies only for MSL which has this strong increasing signal, while for all other ESL components the temporal variability of the global mean is lower than the spatial variability. We have rephrased the sentence into ‘Spatial variations in SLR are considerably lower than the global mean trend’ and we hope it reads better now.

9) Page 7: figures 10-12-> figures S10-S12

Authors: We have modified the manuscript accordingly.

10) “Upgrading existing coastal protection would imply increasing elevations by an average of at least 25 cm by 2050 and by more than 50 cm by 2100”: personally I do not like this statement because it oversimplifies the results and is not representative of the impacts. As the authors themselves claim, the local values may reach 1-2 m, and I am sure that we agree that these are the important numbers.

Authors: Overall we agree with the reviewer, but this is also a matter of style. Often reviewers tend to complain for alarming statements, while others when the authors try to present the findings in a more neutral way. In the paragraph we mention both the global mean and higher local values. Still raising the entire global coastline by 50 cm may not strike as a big issue at first glance but considering the hundreds of thousands km of shoreline the costs (among other practical challenges) are overwhelming. We believe that this is not an important issue but we are willing to make adjustments if the reviewer/editor insists.

11) “On the other hand, working at...” (page 9, SI): I don’t understand the sense of this sentence. Why it refers only to uplift and not to subsidence? Whatever the case, it is in contradiction with the sentence just above.

Authors: Apparently this was a typo and we thank the reviewer for pointing out. We have replaced uplift with subsidence.

Reviewer 2

Review of Vousdoukas et al. “Global probabilistic projections of extreme sea levels”
Reviewer: Sean Vitousek

The paper presents a global-scale estimation of changes in extreme water levels, which contribute to coastal flooding driven by regional sea-level change, waves, tides, and storm surge. The authors examine, for the first time, the effect of non-stationary climate change on the wave, tide, and storm surge patterns on extreme coastal water levels, which should make an excellent addition to the literature. Although, there are a few notable regions which are greatly affected by climatic changes in storminess, such as the North Atlantic and North Pacific. Many regions

are altered insignificantly due to changes in storminess under the RCP 4.5 and 8.5 emission scenarios considered here, and instead are dominated by sea-level rise (SLR), suggesting that earlier works that consider stationary wave climates (Vitousek et al., 2017 – Scientific Reports) may suffice for large-scale coastal hazard assessments.

I was considering writing a paper such as the current manuscript, which considers changes in storminess; however I was glad to see that the authors beat me to it, because I would have not such a thorough analysis as the work presented here. Personally, I am somewhat skeptical of the robustness and utility of non-stationary projections of extreme storminess. Especially, since recent work (Wahl et al., 2017 – Nature Communications) has demonstrated that the uncertainties in extreme water levels can in some cases exceed that of SLR, and therefore trying to estimate non-stationary changes on top of this can be difficult to constrain. Never-the-less, the authors of the current manuscript seem to have gone above and beyond the call of duty in terms of using their modeling efforts to estimate possible changes in storminess, despite their findings of relatively small changes, overall. Ultimately, their findings reinforce the critical role that SLR plays in future hazard projections, as well as a few locations where changes in storminess are important.

Overall, I enjoyed reading the manuscript and I recommend only a few minor revisions.

Authors: We would like to thank Dr Vitousek for his comments and the positive feedback. Regarding the comment on stationary vs non-stationary EVA, we believe that one of the advantages of the latter is that it reduces the uncertainty in the PDF fitting. This is because nsEVA is using longer time series (130 years in the present case) and thus more points to fit the PDF, compared to the 30 years normally applied for stationary EVA (note that the record length has been highlighted as a factor of uncertainty also by Wahl et al. 2017 NCOM). More information is provided in the paper while the code is also open source¹. In any case, the uncertainty from the EVA PDF fitting is included in the analysis of uncertainty shown in Figs 3 and 4 (as part of the uncertainty from climate extremes). We also believe that we expand the discussion on uncertainties compared to previous studies since we consider additional contributions such as ocean model errors, GCM ensemble spread, etc.

Minor revisions:

1) Figure 2: There seems to be some strange behavior between the RCP 4.5 and RCP 8.5 scenarios near western Russia and west Australia where the tendency switches from a positive to a negative contribution from RCP 4.5 to 8.5. I was under the impression that going from 4.5 to 8.5 should increase the tendency, not reverse it. Is there an explanation of what is going on here? Does the fundamental stochastic nature of the climatic extremes realized in these models make this type of analysis difficult?

Authors: This is a very interesting comment. In most cases the RCPs project similar changes, but after a certain point RCP8.5 tends to diverge (Figure 5, SI). West Russia lies in high latitudes where ice effects play an important role, therefore the

estimated trends come with higher uncertainty as our model does not include that variable. This is something that we mention also in the manuscript and is the reason that in general we avoid discussing high latitude areas.

Regarding western Australia, the two RCP in general agree in the general trend which is negative, but it is true that under RCP4.5 some locations show an increase. These locations are close to Indonesia where the topography becomes more complex and local climate effects could justify the deviations. In the figure below we show the projected changes for Australia towards the end of the century, with the lower rows showing the very likely range. It is obvious that the north part of the continent is characterized by high uncertainty, i.e. high GCM ensemble spread (d,e). Note that in Figure 3 of the manuscript points with high uncertainty ($|CV|>1$) are masked, while in the figure below the pattern all the points are shown and the pattern differences in b,c are smaller (but still visible). All the above are related to the wind and pressure fields and how these differ among GCMs. There is no doubt that further interpretation would bring an interesting analysis in the climatological processes of the region and how these are interpreted by GCMs. However, as these are local effects they are considered beyond the scope of the present global study; especially since interpreting further the observed deviations would not be straightforward given the 6 GCMs.

Figure 2. Present contributions of climate extremes to global ESLs (η_{CE}) and projected changes in view of climate change, as well as uncertainty: Maps show the median present-day 100-year η_{CE} (a) and the projected relative changes expressed by the median and the very likely range under RCP4.5 (b, d) and RCP8.5 (c, e) by 2100.

2) Figure 2: This figure shows the absolute change in extreme water level ($\Delta\eta_{\text{ext}}$). A large number of these data points are less than 10-15 cm, while many of the extreme water levels exceed 3 m, suggesting that the percent changes are very small. I personally would be very interested in the percent changes in extreme water level. For example, I may be helpful to have a supplemental figure showing the percent change in extreme water level? Or what percent of locations show less than 5% change, what areas show greater than 10% change. Based on Wahl et al., (2017), I have little doubt that the uncertainty in the stationary extreme water level (in most cases) will be much larger than the non-stationarity. Never-the-less, it is still an important thing to investigate, like the authors have done. Could the uncertainty estimates like in Wahl et al., (2017) be used to estimate the significance of the changes in storminess reported here?

Authors: *The reviewer is starting an interesting discussion of different issues. The reason we have avoided adding relative changes in ESLs is that these are a lot affected by the baseline values. For example 1 m of SLR in a macro-tidal area (max tide >5), exposed to oceanic waves $H_{s,\text{extreme}} > 10$ m, will result in substantially smaller relative change in ESL than at a micro-tidal, sheltered coast. We believe that discussing the results as changes in return periods is more representative since normally coastal protection and other coastal projects are designed on that basis. However, following Reviewer 1's comment 3 we have added versions of (now) Figs 2 and 3 with relative changes in the SL.*

3) Equation 2: $\text{Runup} = 0.2 H_s$. In my opinion, this is quite a bit of an oversimplification, since the wave driven-water level (e.g., setup/runup) will certainly depend on wavelength (or wave period), as in the Stockdon (2006) empirical equation. I see that the authors have used this formulation in a previous report, so it is indeed a published methodology. However, if possible, it may be helpful to examine the influence of wavelength (or wave period), especially to complement the VERY robust approach taken for all of the other water level components.

Authors: *The use of the generic approximation for wave setup comes as an inevitable simplification given the absence of data on the beach profile slope (needed to estimate wave set up) and beach face slope (needed to estimate wave runoff). In terms of hydrodynamics we have all the data required to apply a more elaborate method, and the leading author has substantial experience on swash zone processes²⁻¹², however essential topographic data are missing. The suggestion of the reviewer to add the wave period in the analysis relates to his NSR paper where he applied the wave setup equation for dissipative beaches which is a function of wave height and wavelength. This is definitely not wrong, but in our case would mean assuming that all beaches worldwide are dissipative. Wave runoff consists of wave setup and swash and dissipative beaches tend to have higher wave setup due to saturated surf zones, while on reflective ones swash is dominant. Therefore, in our opinion there is no 'correct' answer to this issue, but most likely, advances in remote sensing will provide us soon the high resolution*

coastal relief data that are currently missing (i.e. shoaling zone and beach-face slopes).

In the absence of proper data we have applied the generic approximation for wave setup as we also find that it is more compatible with the current state of the art in flood and risk assessments; for which we hope that our dataset will be useful. Most studies apply the static inundation approach which anyway overestimates flood extents¹³. The wave runup height is not persistent during an extreme event and its use would result in further overestimation. Even in studies which apply hydrological models for coastal inundation¹⁴, the time steps of the simulations are not small enough to resolve wave oscillations and wave runup would be an overprediction of the forcing water level. For that reason wave setup, being a slower, more persistent episodic elevation of the sea level than wave runup, was chosen.

4) Page 5: Local changes in the mean high tide water level: I think the discussion on the changes in the tide due to SLR should be shortened even further in the main text, as in my opinion it would be completely insignificant in comparison to the other factors (and uncertainties) affecting extreme water level. If I were conducting such a study, I would have guessed that the effect of SLR on the astronomic tide would be completely insignificant (perhaps except in shallow estuaries), and thus I never would have conducted such an analysis. However, I commend the authors' diligence to actually verify this for the rest of us.

Authors: In our opinion the effect of SLR on tides remains largely unknown since recent studies have shown different effects when the shoreline is considered fixed, or allows to retreat with SLR¹⁵. Given that it has little effect in our projections we dedicate only 1 paragraph in discussing the results. However, not discussing at all does not seem appropriate since the fact that we consider changes in tides is mentioned in the abstract and methods.

5) Figure 3: I am a bit unclear on what is meant by variance and the sources of it here. I would appreciate some additional discussion and examples on how this is estimated.

Authors: We agree with the reviewer that our analysis on the relative contributions to the uncertainty was not well represented in the manuscript. We hope this issue has been solved in the revision where we have provided additional information on the variance and the analysis on the sources of uncertainty.

(As a side note: Why is the variance for Delta eta CE constant for RCP4.5? but not for RCP 8.5?)

Authors: A careful look at the figure will show that the variance under RCP4.5 is also increasing, even though not as much as under RCP8.5. This is probably related to the GCM ensemble spread which is smaller under the moderate emission mitigation scenario.

6) Figures 1,2,5: In our paper (Vitousek et al., 2017), we presented our global-scale estimates of coastal hazards using continuous color plots (like in Figure S9) rather than at discrete points along the coastline. This approach seemed more intuitive to us since, we were considering continuous oceanic variables (despite their coastal relevance), and the clipping of those variables to coastal points seemed somewhat arbitrary and furthermore it may result in some under representation of small island nations. I am not suggesting that the authors change this approach, however, I wonder if they have a compelling reason/motivation for keeping the results only near the (major) coastlines? We debated these different approaches with our reviewers before ultimately sticking with the continuous maps.

Authors: Plotting maps instead of scatter plots would definitely improve aesthetically the figures, however in LISCOAST we focused only on coastal points, and information along the entire domain is stored only in lower temporal and spatial resolution. That was a conscious decision during the design of the simulations. Given the temporal and spatial scales of the study, as well as the number of RCPs, GCMs, models we had to prioritize our use of computational and storage resources. Therefore we decided to give priority on having high resolution along the coast which is the main scope of LISCOAST, with the cost of neglecting off-shore locations.

7) Figures S5 and S13: I really like these analyses and figures a lot. To me, they are more informative than Figures 2 and 5, respectively. Thus, I believe they may be worth migrating to the main article rather than in the supplemental materials. However, I am happy with whatever the authors/editors decide.

Authors: We thank his reviewer for the positive feedback and the suggestion and we have moved the figures to the main manuscript.

Reviewer 3

This is an interesting paper that deals with an important and timely topic. The authors employ a probabilistic framework to assess future changes in the different components of extreme sea levels (i.e. mean sea level, tides, storm surges, and waves). Their approach allows the authors to quantify the contributions of individual drivers and the associated spatial variability. I think the presented results are significant and relevant to the broad readership of Nature Communications. However, there are a few aspects that need more attention/revision, before the manuscript can be recommended for publication.

General comments:

- A paper was recently published by Rasmussen et al. "Extreme sea level implications of 1.5 °C, 2.0 °C, and 2.5 °C temperature stabilization targets in the 21st and 22nd century", <https://doi.org/10.1088/1748-9326/aaac87>. The paper was probably not available at the time the authors submitted their manuscript, but it is now, and since it deals with a similar topic it would be important to discuss the results from this present manuscript in light of the results of this other study. I believe that this

manuscript is different enough and has new significant findings still warranting its publication through a high-level outlet.

Authors: It is true that the suggested paper was published after our submission and we are thankful to the reviewer for pointing it out. We have included the paper in the revision.

- A bit more explanation for using $0.2H_s$ as an approximation for the wave setup would be interesting. In their paper, Vitousek et al. (2017) use a slightly different approach. It would be important to understand how sensitive the results are to different wave setup approximations.

Authors: This comment relates to comment 3 from Dr Vitousek (Reviewer 2) and is discussed in detail in our reply there.

- In the supplementary material I was a bit confused by the discussion on the importance of TCs and how they are incorporated. First, for the historical period, the authors correctly state that TC storm surges are likely underestimated when using only the reanalysis data as boundary conditions, and hence decided to include observed TC information from various data bases as additional forcing. Then, for future simulations, it is stated on page 6 that relative changes in TC storm surges should be captured by the GCMs. Further down, on page 10, the fact that GCMs poorly resolve TCs is then listed as a limiting factor in the studies. I think this is a crucial aspect and needs more/better explanation and consistency.

Authors: This comment relates to comment 4 from Reviewer 1 and we encourage the reviewer to check our previous reply. At the same time we acknowledge that aspects of the methodology related to TCs could be better explained and we have improved the manuscript on that direction.

Detailed comments:

- In the current version the authors introduce their definition of ESL at the beginning of the supplementary material (i.e. sum of MSL, tide, surge, wave setup). Given that the entire study is built around this definition it should also appear prominently in the main text.

Authors: We have added the ESL definition in the last paragraph of the introduction.

- P.2 first para. Saying that only “low-latitude” coastal zones are affected by tropical cyclones is a bit misleading and ill-defined. I am thinking about events travelling up the US east coast affecting areas well beyond 35 or 40 degrees latitude.

Authors: The comment is fair and we have rephrased the sentence which now reads ‘ESLs are exacerbated from tropical cyclones, which significantly intensify wind-waves and storm surge^{16,17}.’

- P4. It might be clearer to say decrease and then use -10 to -32 cm. Technically it's not wrong how it is done now, but I was a bit puzzled first.

Authors: We have modified the whole paragraph accordingly.

- P.7 It should be "Figures S10-S12"

Authors: We have corrected accordingly, as this was also pointed by reviewer 1.

References

- 1 Mentaschi, L. *et al.* Non-stationary Extreme Value Analysis: a simplified approach for Earth science applications. *Hydrol. Earth Syst. Sci. Discuss.* **2016**, 1-38, doi:10.5194/hess-2016-65 (2016).
- 2 Vousdoukas, M. I. Observations of wave run-up and groundwater seepage line motions on a reflective-to-intermediate, meso-tidal beach. *Mar. Geol.* **350**, 52-70, doi:<http://dx.doi.org/10.1016/j.margeo.2014.02.005> (2014).
- 3 Vousdoukas, M. I. *et al.* The role of combined laser scanning and video techniques in monitoring wave-by-wave swash zone processes. *Coastal Eng.* **83**, 150-165, doi:<http://dx.doi.org/10.1016/j.coastaleng.2013.10.013> (2014).
- 4 Vousdoukas, M. I. *et al.* in *7th International Conference on Coastal Dynamics* 11 (Bordeaux, France, 2013).
- 5 Almeida, L. P., Vousdoukas, M. I., Ferreira, Ó., Rodrigues, B. A. & Matias, A. Thresholds for storm impacts on an exposed sandy coastal area in southern Portugal. *Geomorphology* **143-144**, 3-12, doi:10.1016/j.geomorph.2011.04.047 (2012).
- 6 Vousdoukas, M. I., Almeida, L. P. & Ferreira, Ó. Beach erosion and recovery during consecutive storms at a steep-sloping, meso-tidal beach. *Earth Surf. Processes Landforms* **37**, 583-691, doi:10.1002/esp.2264 (2012).
- 7 Vousdoukas, M. I. Erosion/accretion and multiple beach cusp systems on a meso-tidal, steeply-sloping beach. *Geomorphology* **141-142**, 34-46, doi:10.1016/j.geomorph.2011.12.003 (2012).
- 8 Schimmels, S., Vousdoukas, M. I., Oumeraci, H. & Wziatek, D. in *33rd International Conference on Coastal Engineering* (Santander, Spain, 2012).
- 9 Vousdoukas, M. I., Ferreira, O., Almeida, L. P. & Pacheco, A. Toward reliable storm-hazard forecasts: XBeach calibration and its potential application in an operational early-warning system. *Ocean Dyn.* **62**, 1001-1015, doi:10.1007/s10236-012-0544-6 (2012).
- 10 Vousdoukas, M. I., Wziatek, D. & Almeida, L. P. Coastal vulnerability assessment based on video wave run-up observations at a mesotidal, steep-sloped beach. *Ocean Dyn.* **62**, 123-137, doi:10.1007/s10236-011-0480-x (2012).
- 11 Vousdoukas, M. I., Almeida, L. P. & Ferreira, O. Modelling storm-induced beach morphological change in a meso-tidal, reflective beach using XBeach. *J. Coast. Res.*, 1916-1920 (2011).

- 12 Vousdoukas, M. I., Velegrakis, A. F., Dimou, K., Zervakis, V. & Conley, D. C. Wave run-up observations in microtidal, sediment-starved pocket beaches of the Eastern Mediterranean. *Journal of Marine Systems* **78**, S37-S47 (2009).
- 13 Vousdoukas, M. I. *et al.* Developments in large-scale coastal flood hazard mapping. *Natural Hazards and Earth System Science* **16**, 1841-1853, doi:10.5194/nhess-16-1841-2016 (2016).
- 14 Ramirez, J. A., Lichter, M., Coulthard, T. J. & Skinner, C. Hyper-resolution mapping of regional storm surge and tide flooding: comparison of static and dynamic models. *Nat. Hazards* **82**, 571-590, doi:10.1007/s11069-016-2198-z (2016).
- 15 Du, J. *et al.* Tidal Response to Sea-Level Rise in Different Types of Estuaries: The Importance of Length, Bathymetry, and Geometry. *Geophys. Res. Lett.* **45**, 227-235, doi:10.1002/2017GL075963 (2018).
- 16 Little, C. M. *et al.* Joint projections of US East Coast sea level and storm surge. *Nature Clim. Change* **5**, 1114-1120, doi:10.1038/nclimate2801 <http://www.nature.com/nclimate/journal/v5/n12/abs/nclimate2801.html#supplementary-information> (2015).
- 17 Peduzzi, P. *et al.* Global trends in tropical cyclone risk. *Nature Clim. Change* **2**, 289-294, doi:<http://www.nature.com/nclimate/journal/v2/n4/abs/nclimate1410.html#supplementary-information> (2012).

REVIEWERS' COMMENTS:

Reviewer #1 (Remarks to the Author):

I want to thank the authors for their detailed responses to my queries. I believe that the manuscript is ready for publication. I just have one additional comment that the authors may want to consider in their final version.

- regarding my former major point 1: there is a recently submitted manuscript that discusses the assumption of independence of different contributors, in this case applied to mean sea level rise. It could be worth to cite the limitations pointed out here. The manuscript can be found at <https://eartharxiv.org/uvw3s/>

Reviewer #2 (Remarks to the Author):

I am satisfied with the authors' responses to my comments and their revisions to the manuscript. I have only a few minor recommendations before the manuscript is ready for publication.

Minor revision #1:

The "Limitations" section is a welcome addition to the paper.

To this section I would recommending adding a discussion on the limitations of the method used to estimate wave setup from wave height. Here, the authors can discuss some of the limitations associated with using $\text{Setup} = 0.2 * H_s$ and the lack of global information on beach slope which lead to the decision to use this approach. This section can mostly repeat our per-existing discussion from your response letter to my earlier recommendations (and those from reviewer #3, which reiterated my concerns).

Also, I would like to call the authors' attention to a recent paper on this subject:

Melet, A., Meyssignac, B., Almar, R., & Le Cozannet, G. (2018). Under-estimated wave contribution to coastal sea-level rise. *Nature Climate Change*, 1.

who surveyed a few different methods to calculating setup + swash and applied them on a global scale. The discussion on which empirical runup/setup equation to use to apply on a global scale is worth repeating here, if only briefly.

Specific Comments:

Line 50: eta_ce -> capitalization needed to be consistent with other instances.

Figure 1 is a welcome addition to the paper. Perhaps some color would help make the figure look even better.

Line 151: Under the buissess as usual scenario, the highest SLR ...
(comma needed instead of period)

Line 216: "For future estimates, eta_ce uncertainty is enhanced by the one in predicting future atmospheric conditions (i.e. inter-GCM variability)." This is a bit unclear. Consider rewording.

Line 267: Is there a section 0? I am not sure where this is.

Reviewer #3 (Remarks to the Author):

The authors have responded well to all my previous comments and revised the manuscript accordingly. I have no further concerns and congratulate the authors on a really nice and very important paper.

Sincerely,
Thomas Wahl

REVIEWERS COMMENTS

Authors: *We have responded to all the comments and suggestions from the reviewers and the editor. More specifically we have modified all figures and the title according to the editor's suggestions. We have applied the journal's style and made sure that all points in the checklist are covered. Our responses to suggestions from the reviewers are discussed below.*

Reviewer #1

I want to thank the authors for their detailed responses to my queries. I believe that the manuscript is ready for publication. I just have one additional comment that the authors may want to consider in their final version.

- regarding my former major point 1: there is a recently submitted manuscript that discusses the assumption of independence of different contributors, in this case applied to mean sea level rise. It could be worth to cite the limitations pointed out here. The manuscript can be found at <https://eartharxiv.org/uvw3s/>

Authors: *We would like to thank the reviewer and we have considered the suggested study in our discussion*

Reviewer #2

I am satisfied with the authors' responses to my comments and their revisions to the manuscript. I have only a few minor recommendations before the manuscript is ready for publication.

Minor revision #1:

The "Limitations" section is a welcome addition to the paper.

To this section I would recommending adding a discussion on the limitations of the method used to estimate wave setup from wave height. Here, the authors can discuss some of the limitations associated with using $Setup = 0.2 * H_s$ and the lack of global information on beach slope which lead to the decision to use this approach. This section can mostly repeat our per-existing discussion from your response letter to my earlier recommendations (and those from reviewer #3, which reiterated my concerns).

Also, I would like to call the authors' attention to a recent paper on this subject:

Melet, A., Meyssignac, B., Almar, R., & Le Cozannet, G. (2018). Under-estimated wave contribution to coastal sea-level rise. *Nature Climate Change*, 1.

who surveyed a few different methods to calculating setup + swash and applied them on a global scale. The discussion on which emperical runup/setup equation to use to apply on a global scale is worth repeating here, if only briefly.

Authors: *We have added the discussion on wave setup and we have considered the suggested study in our discussion*

Specific Comments:

Line 50: eta_ce -> capitalization needed to be consistent with other instances.

Authors: *We have corrected the text accordingly.*

Figure 1 is a welcome addition to the paper. Perhaps some color would help make the figure look even better.

Authors: *We believe that this is a useful but less important comment and we would prefer not to change the figure. The reason is that we tried to add colors in previous versions and the aesthetic result was inferior than the present.*

Line 151: Under the buisness as usual scenario, the highest SLR ...

(comma needed instead of period)

Authors: *We have corrected the manuscript accordingly.*

Line 216: "For future estimates, eta_ce uncertainty is enhanced by the one in predicting future atmospheric conditions (i.e. inter-GCM variability)." This is a bit unclear. Consider rewording.

Authors: *We have rephrased the sentence and we hope it reads better now.*

Line 267: Is there a section 0? I am not sure where this is.

Authors: *Apparently this is a typo and has been corrected.*

Reviewer #3

The authors have responded well to all my previous comments and revised the manuscript accordingly. I have no further concerns and congratulate the authors on a really nice and very important paper.

Sincerely,

Thomas Wahl

Authors: *We have like to thank Dr Wahl for his constructive review and comments.*